# Kinetically matched C−N coupling toward efficient urea electrosynthesis enabled on copper single-atom alloy

Mengqiu Xu[1,4], Fangfang Wu[2,4], Ye Zhang[1], Yuanhui Yao[1], Genping Zhu[1], Xiaoyu Li[1], Liang Chen ●[1] ✉, Gan Jia[1], Xiaohong Wu ●[3] ✉, Youju Huang ●[1], Peng Gao ●[1] ✉ & Wei Ye ●[1] ✉

Chemical C−N coupling from $CO_2$ and $NO_3^-$, driven by renewable electricity, toward urea synthesis is an appealing alternative for Bosch−Meiser urea production. However, the unmatched kinetics in $CO_2$ and $NO_3^-$ reduction reactions and the complexity of C- and N-species involved in the co-reduction render the challenge of C−N coupling, leading to the low urea yield rate and Faradaic efficiency. Here, we report a single-atom copper-alloyed Pd catalyst ($Pd_4Cu_1$) that can achieve highly efficient C−N coupling toward urea electrosynthesis. The reduction kinetics of $CO_2$ and $NO_3^-$ is regulated and matched by steering Cu doping level and $Pd_4Cu_1/FeNi(OH)_2$ interface. Charge-polarized $Pd^{\delta-}$-$Cu^{\delta+}$ dual-sites stabilize the key *CO and *$NH_2$ intermediates to promote C−N coupling. The synthesized $Pd_4Cu_1$-$FeNi(OH)_2$ composite catalyst achieves a urea yield rate of 436.9 mmol $g_{cat.}^{-1}$ $h^{-1}$ and Faradaic efficiency of 66.4%, as well as a long cycling stability of 1000 h. In-situ spectroscopic results and theoretical calculation reveal that atomically dispersed Cu in Pd lattice promotes the deep reduction of $NO_3^-$ to *$NH_2$, and the Pd-Cu dual-sites lower the energy barrier of the pivotal C−N coupling between *$NH_2$ and *CO.

Urea ($CO(NH_2)_2$) is a vital chemical fertilizer in modern society, which greatly promotes the development of agriculture and contributes to the rapid growth of world's population[1–3]. Industrial urea production relies on the Bosch−Meiser process, in which carbon dioxide ($CO_2$) and ammonia ($NH_3$) are thermochemically coupled operated at elevated temperatures (~200 °C) and high pressures (~210 bar)[4]. Approximately 80% of industrial $NH_3$ produced by the Haber−Bosch process is fed for the urea production[5]. Consequently, the harsh conditions in urea synthesis consume substantial fossil fuels, and which leads to serious $CO_2$ release. Urea electrosynthesis from $CO_2$ and nitrogenous compounds is an attractive alternative approach by taking advantage of the in situ generated C- and N-intermediates. As the electrolytic reactions can be carried out at room temperature and atmospheric pressure, the energy efficiency can be greatly improved. Nonetheless, restricted by the inert N≡N bond (bond energy of 941 kJ $mol^{-1}$) and low solubility of $N_2$ in aqueous electrolytes, the urea electrosynthesis from $CO_2$ and $N_2$ delivers low urea yield rates (typically <5 mmol $g_{cat.}^{-1}$ $h^{-1}$) and urea Faradaic efficiency (FE, <20%)[6–8]. The nitrate ions ($NO_3^-$) reduction reaction ($NO_3RR$) is easier than $N_2$ reduction, due to the lower N=O bond energy (206 kJ $mol^{-1}$) and much higher solubility of $NO_3^-$ [9,10]. Nitrate ions are also an abundant feedstock, mainly come from industrial wastewater, chemical fertilizers, and livestock excrement, which may serve as ideal candidates for the C−N coupling[11].

[1]College of Material, Chemistry and Chemical Engineering, Key Laboratory of Organosilicon Chemistry and Material Technology, Ministry of Education, Hangzhou Normal University, 311121 Hangzhou, Zhejiang, China. [2]College of Materials Science and Engineering, Zhejiang University of Technology, 310014 Hangzhou, Zhejiang, China. [3]School of Chemistry and Chemical Engineering, Harbin Institute of Technology, 150001 Harbin, Heilongjiang, P. R. China. [4]These authors contributed equally: Mengqiu Xu, Fangfang Wu. ✉e-mail: liang_chen@hznu.edu.cn; wuxiaohong@hit.edu.cn; gaopeng@hrbeu.edu.cn; yewei@hznu.edu.cn

Urea yield rate and urea FE in urea electrosynthesis from $CO_2$ and $NO_3^-$ are still insufficient compared to the thresholds of economic viability predicted by techno-economic assessments. An efficient C−N coupling electrocatalyst should possess the following features. First, the matched kinetics of $NO_3RR$ and $CO_2$ reduction reaction ($CO_2RR$) is the prerequisite to boost urea yield rate and FE (see Supplementary Fig. 1). Second, the adjacent dual-sites are required to stabilize C- and N-intermediates, respectively and lower the energy barrier of C−N coupling. Third, the possible by-products should be effectively restrained to ensure high urea FE as varieties of C- and N-species are inevitably involved in the co-reduction process (e.g., CO, $CH_4$, $CH_3OH$ and HCOOH in $CO_2RR$, $NO_2^-$, $NH_3$, $NH_2OH$, $N_2$ in $NO_3RR$)[12–16]. Taken these regards, electrocatalyst with tunable dual-sites is an ideal choice to induce the formation and stabilize the pivotal C- and N-intermediates (*CO and *$NH_2$, * denotes the active site) for C−N coupling[15,17]. As *CO is electron deficient and *$NH_2$ is electron efficient, constructing $M_1^{\delta-}$-$M_2^{\delta+}$ (e.g., $M_1$ = Pd, $M_2$ = Cu) type dual-sites with charge polarization seems to be effective for stabilization of the key intermediates.

Here, we design the charge-polarized $Pd^{\delta-}$-$Cu^{\delta+}$ dual-sites in copper single-atom alloy toward efficient electrochemical C−N coupling. Atomically dispersed Cu atoms in Pd lattice accelerate $NO_3RR$ by promoting the deep reduction of $NO_2^-$ to *$NH_2$. Meanwhile, the reduction of $CO_2$ to CO is also strengthened, while the desorption process of *CO is restrained on Cu single-atom alloy. Therefore, the kinetics of $NO_3RR$ and $CO_2RR$ is well matched with N- and C-intermediates yield rate ratio of 1.5, which is close to the stoichiometric ratio (2:1) in urea. In situ Raman spectroscopic characterizations combined with theoretical calculation reveal that $Pd^{\delta-}$-$Cu^{\delta+}$ dual-sites stabilize the two key intermediates (*CO and *$NH_2$) for C−N coupling, respectively. Benefitting from the matched kinetics and charge-polarized dual-sites in Cu single-atom alloy, $Pd_4Cu_1$-$Ni(OH)_2$ catalyst delivers urea yield rate of 60.4 mmol $g_{cat.}^{-1}$ $h^{-1}$ and urea FE of 64.4% in gas diffusion electrode (GDE, catalyst loading: 0.1 mg $cm^{-2}$). Further optimizing the carrier with Fe-doping in $Ni(OH)_2$ to accelerate water dissociation and improve the yield rates of N- and C-intermediates, the $Pd_4Cu_1$-$FeNi(OH)_2$ composite catalyst delivers the urea yield rate of 436.9 mmol $g_{cat.}^{-1}$ $h^{-1}$ and FE of 66.4%, together with the high catalytic stability up to 1000 h in GDE.

## Results

### Synthesis and structural characterization of electrocatalysts

Atomic dispersion of Cu in Pd lattice was synthesized by co-reduction of $PdCl_4^{2-}$ and $Cu^{2+}$ with $NaBH_4$ as a reducing agent. Ultrathin layered α-$Ni(OH)_2$ nanosheets were employed to accelerate water splitting to produce more active hydrogen atoms and used as catalyst carrier (Supplementary Fig. 2). The synthetic process of the composite electrocatalyst is demonstrated in Supplementary Fig. 3. Cu doping level in Pd host was controlled by regulating the molar ratios of Pd:Cu precursors. As shown in Supplementary Table 1, the molar ratios of Pd:Cu in the as-synthesized products determined by inductively coupled plasma-mass spectrometry (ICP-MS) are consistent with these of Pd:Cu precursors. Therefore, the samples are denoted as $Pd_xCu_1$-$Ni(OH)_2$ (x = 1, 2, 3, 4, 5, 6). Among which, solid solution phase alloy, i.e., $Pd_1Cu_1$ clusters, are formed. Atomic dispersion of Cu atoms in Pd lattice is formed by decreasing Cu doping level to Pd:Cu ratio of 4:1[18]. Powder X-ray diffraction (XRD) patterns of the composite samples only display the diffraction patterns of α-$Ni(OH)_2$, without face-centered cubic (fcc) phase Pd/Cu (Supplementary Fig. 4). Transmission electron microscopic (TEM, Supplementary Fig. 5) characterization demonstrates that the metal clusters are anchored on $Ni(OH)_2$ nanosheets. Taken $Pd_4Cu_1$-$Ni(OH)_2$ as an example, aberration-corrected high-angle annular dark-field scanning TEM (HAADF-STEM, Fig. 1a and TEM image in Supplementary Fig. 6) image shows that $Pd_4Cu_1$ clusters with average size of 3.5 ± 0.1 nm are uniformly distributed on α-$Ni(OH)_2$

nanosheets. High-resolution HAADF-STEM (Fig. 1b) image indicates the spherical $Pd_4Cu_1$ nanoparticles, where the lattice distance of 0.22 nm can be attributed to (111) plane of fcc Pd/Cu.

The elemental mapping profile (Fig. 1c) indicates a uniform distribution of Pd and Cu across $Pd_4Cu_1$ cluster, manifesting a uniform Cu doping in Pd lattice[19]. Then, X-ray photoelectron spectroscopic (XPS, Supplementary Fig. 7) result confirms the existence of Pd and Cu with molar ratio approaching 4:1, consistent with ICP-MS result. As shown in Fig. 1d, the binding energy of Cu $2p_{3/2}$ for metallic Cu shifts from 932.3 eV to higher value of 932.6 eV for $Pd_1Cu_1$ and $Pd_4Cu_1$ clusters. The result indicates that electrons are denoted from Cu to adjacent Pd atoms, due to the larger electronegativity of Pd atoms than Cu, leading to the formation of charge-polarized $Pd^{\delta-}$-$Cu^{\delta+}$ dual-sites[20,21]. In addition, a satellite peak around 941.4 eV can be assigned to $Cu^{2+}$ in $Pd_1Cu_1$-$Ni(OH)_2$ sample[22].

To decode the exact fine structure of copper single-atom alloy structure, $Pd_4Cu_1$-$Ni(OH)_2$ was characterized by synchrotron radiation-based X-ray absorption fine structure (XAFS) spectroscopy. Figure 1e shows Cu K-edge X-ray absorption near edge structure (XANES) spectra of $Pd_4Cu_1$-$Ni(OH)_2$ in reference with CuO and Cu foil. The intensity (the insert in Fig. 1e) of Cu K-edge between 8975 and 8995 eV for $Pd_4Cu_1$-$Ni(OH)_2$ sample is slightly lower than that of Cu foil. It manifests that the valence of $Cu^{\delta+}$ in $Pd_4Cu_1$ is approaching $Cu^0$ but slightly higher than $Cu^0$, confirming the charge polarization ($Cu^{\delta+}$ → $Pd^{\delta-}$) between Cu and adjacent Pd atoms[23]. Cu extended XAFS (EXAFS) spectra were obtained through a Fourier transformation of Cu K-edge spectra (Fig. 1f). The fine crystalline structure is confirmed by fitting the $k^3$-weighted Fourier transformed EXAFS spectra (Fig. 1g and Supplementary Fig. 8). In contrast with Cu foil, Cu−Cu bond is absent in $Pd_4Cu_1$-$Ni(OH)_2$ sample. Cu−Pd bond (2.61 Å) is resolved in the first shell with a coordination number (CN, Supplementary Table 2) of 10.7, verifying the isolated Cu atoms in Pd lattice[24,25]. Besides, Cu−O bond (2.05 Å, CN = 3.1) is also observed in $Pd_4Cu_1$-$Ni(OH)_2$ sample, revealing partial oxidation of Cu atoms[18]. Then, wavelet transforms (WT) analysis of the Cu K-edge EXAFS oscillations of $Pd_4Cu_1$-$Ni(OH)_2$ sample was performed in reference with CuO, Cu foil. Two dimensional contour maps of $Pd_4Cu_1$-$Ni(OH)_2$ in Fig. 1h resolve Pd−Cu bond, while Cu−Cu bond is absent determined by the wave vector number (**k**). The fine structure of Pd was also resolved by XAFS (Supplementary Fig. 9). Putting together the above results, we come to a conclusion that Cu is atomically dispersed in Pd lattice, namely Cu single-atom alloy.

### Evaluation of catalytic performance

Urea electrosynthesis test was carried out in an H-type cell at room temperature with gaseous $CO_2$ and $KNO_3$ as C- and N-sources, respectively. Linear sweep voltammetry (LSV) test was initially carried out to evaluate current response for $Pd_4Cu_1$-$Ni(OH)_2$ sample. As shown in Fig. 2a, the current densities are in the sequence of $I(KNO_3) > I(KNO_3 + KHCO_3) > I(KHCO_3 + CO_2) > I(KNO_3 + KHCO_3 + CO_2)$. The results indicate that the co-reduction of $NO_3^-$ and $CO_2$ toward C−N coupling delivers lower current density than that of solo $NO_3RR$ or $CO_2RR$, suggesting $NO_3RR$, $CO_2RR$ and the competing hydrogen evolution reaction are effectively suppressed in the co-electrolysis[3,12]. Then, we screened the optimal urea yield rate and FE at −0.5 V versus reversible hydrogen electrode (RHE) over $Pd_xCu_1$-$Ni(OH)_2$ composite catalysts in H-type cell, in contrast with bare $Ni(OH)_2$ nanosheets or Pd-$Ni(OH)_2$ sample. The loading amount of $Pd_xCu_1$ in the sample toward urea electrosynthesis was firstly optimized (Supplementary Fig. 10). The produced amount of urea in the electrolyte was spectrophotometrically quantified using diacetyl monoxime as chromogenic reagent (Supplementary Fig. 11)[3]. As shown in Fig. 2b, urea yield rates and urea FEs all show a volcano-shape variation trend with Pd:Cu molar ratios ($Pd_xCu_1$, x = 1−6). Notably, $Pd_xCu_1$-$Ni(OH)_2$ (x = 1−6) composite electrocatalysts all deliver higher urea electrosynthesis performance than that of bare $Ni(OH)_2$ nanosheets (0.9 mmol $g_{cat.}^{-1}$ $h^{-1}$, 1.4%) and Pd-

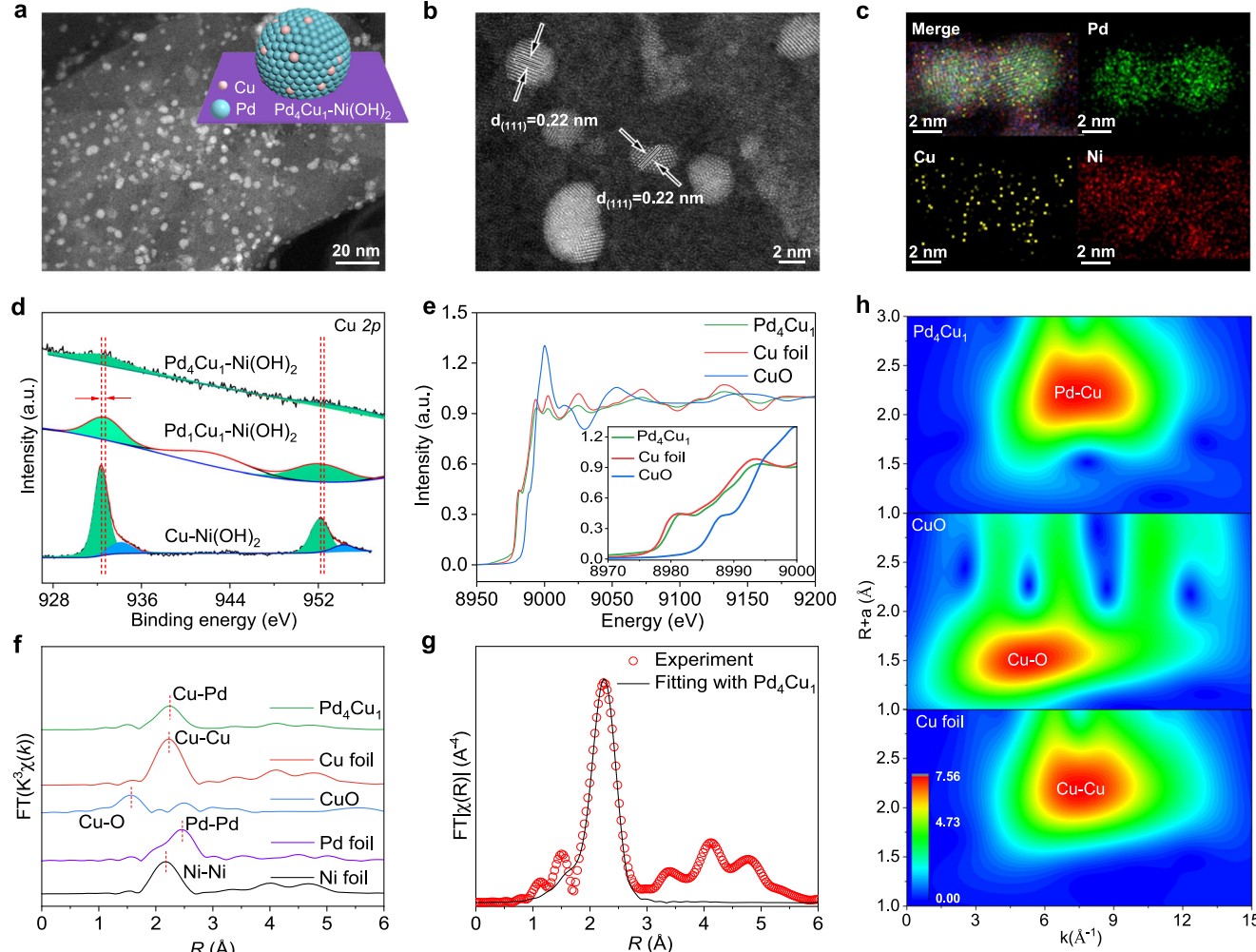

**Fig. 1 | Characterization of Pd₄Cu₁-Ni(OH)₂ sample. a** HAADF-STEM image, **b** high-resolution HAADF-STEM image, **c** EDS elemental mapping profile of Pd₄Cu₁-Ni(OH)₂ composite structure. **d** Cu 2p spectra of Pd₄Cu₁-Ni(OH)₂, Pd₁Cu₁-Ni(OH)₂ and Cu-Ni(OH)₂. **e** Normalized Cu K-edge XANES spectra of Pd₄Cu₁ clusters in reference with Cu foil and CuO, **f** $k^3$-weighted Fourier-transform Cu K-edge, Pd K-edge and Ni K-edge EXAFS spectra, **g** the experimental Cu K-edge EXAFS spectrum (red circle) and the fitting curve (black line) of Pd₄Cu₁. **h** Wavelet transforms of the $k^2$-weighted Cu K-edge EXAFS signals for the high-coordination shells in reference with Cu foil and CuO. The inset in **a** shows schematic diagram of Pd₄Cu₁-Ni(OH)₂.

Ni(OH)₂ (2.3 mmol $g_{cat.}^{-1}$ h⁻¹, 6.6%). The optimal urea yield rate and urea FE are 18.8 mmol $g_{cat.}^{-1}$ h⁻¹ and 76.2% achieved on Pd₄Cu₁-Ni(OH)₂ sample with urea partial current density of 0.68 mA cm⁻² (Supplementary Fig. 14a). Urea yield rates are about 20.9- and 8.2-fold higher than that of bare Ni(OH)₂ and Pd-Ni(OH)₂ counterparts, respectively. The above results indicate that alloying Cu single-atoms in Pd lattice really boosts urea electrosynthesis performance (Supplementary Fig. 12).

Then, potential-dependent urea yield rates and FEs of Pd₄Cu₁-Ni(OH)₂ in H-type cell were also assessed (Supplementary Fig. 13). As indicated in Fig. 2c, urea yield rates are 3.4, 1.5, 3.2, 3.8, 18.8 and 9.2 mmol $g_{cat.}^{-1}$ h⁻¹ at −0.1, −0.2, −0.3, −0.4, −0.5 and −0.6 V, respectively. Correspondingly, urea FEs are 14.0%, 14.0%, 16.0%, 31.1%, 76.2% and 33.8%. To exclude the impact of NO₂⁻ in the electrolyte derived from NO₃RR on urea determination, the produced amount of urea in the electrolyte was also quantified through spectrophotometric method with urease and ¹H-NMR spectroscopy (Supplementary Figs. 15–17)[26]. In addition, N- and C-selectivity reaches 88.6% and 96.1% (Supplementary Fig. 18) in urea electrosynthesis at −0.5 V, respectively. ¹⁵N isotope labeling experiments (¹⁵NO₃⁻ as feeding) were carried out to further confirm the produced urea was rooted from the C−N coupling of NO₃⁻ and CO₂ (Supplementary Figs. 19 and 20)[9]. To show the unique promotion role of Cu single-atom alloy, we also screened the transition

metals in single-atom alloys (Pd₄X₁, X=Fe, Co, Ni, Cu, Zn) for C−N coupling, and the result indicates the best choice of Cu (Supplementary Figs. 21 and 22).

Urea electrosynthesis was further assessed in commercial GDE (Supplementary Fig. 23) to improve mass transfer of CO₂. Figure 2d shows potential-dependent urea yield rates and FEs of Pd₄Cu₁-Ni(OH)₂ in GDE with CO₂ flow rate of 20 mL min⁻¹. Urea yield rates are 6.2, 7.2, 9.9, 13.5, 60.4 and 47.3 mmol $g_{cat.}^{-1}$ h⁻¹ at −0.1, −0.2, −0.3, −0.4, −0.5 and −0.6 V, respectively, which are obviously higher than that in H-type cell. Urea FEs are 19.6%, 27.7%, 22.5%, 39.6%, 64.4% and 54.5% between −0.1 and −0.6 V. Urea partial current density in GDE increases to 2.3 mA cm⁻² at −0.5 V (Supplementary Fig. 14b, c). The optimal urea yield rate (60.4 mmol $g_{cat.}^{-1}$ h⁻¹) and FE (64.4%) at −0.5 V exceed the current state-of-the-art electrocatalysts as summarized in Supplementary Table 3.

Apart from urea yield rate and FE, cycling stability is another important parameter in the catalyst evaluation. As shown in Fig. 2e, urea partial current density ($j_{urea}$) in H-type cell stabilizes in the initial 40 h, and then slightly declines in the following 60 h. In addition, urea yield rate slightly declines to 12.9 mmol $g_{cat.}^{-1}$ h⁻¹ at 100 h with retention of 68.7%. After durability test (100 h), Pd₄Cu₁ still sustains cluster structure on Ni(OH)₂ nanosheets without obvious size changes,

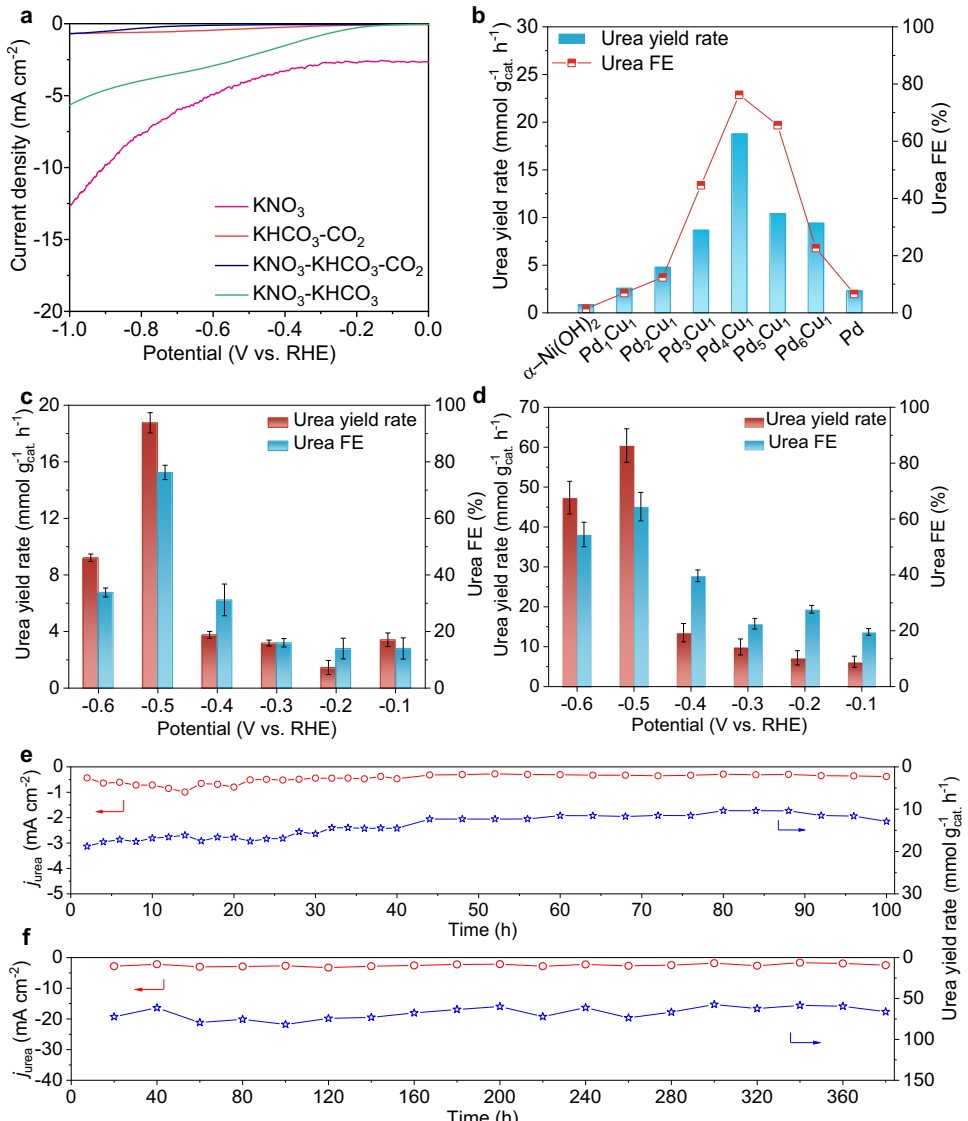

**Fig. 2 | Urea electrosynthesis performance. a** LSV curves of $Pd_4Cu_1$-$Ni(OH)_2$ recorded in the mixture of 0.1 M $KHCO_3$ + 0.1 M $KNO_3$ (pH=8.4) under $CO_2$ flow in reference with that in 0.1 M $KNO_3$, 0.1 M $KHCO_3$ + $CO_2$, 0.1 M $KNO_3$ + 0.1 M $KHCO_3$. **b** Screening electrocatalysts toward urea electrosynthesis with $Pd_xCu_1$-$Ni(OH)_2$ composite samples. Potential-dependent urea yield rates and FEs of $Pd_4Cu_1$-$Ni(OH)_2$ in **c** H-type cell and **d** GDE with catalyst loading: 0.1 mg cm$^{-2}$. Cycling stability of $Pd_4Cu_1$-$Ni(OH)_2$ catalyst in urea electrosynthesis assessed **e** in H-type cell and **f** in GDE. **c, d** Error bars in accordance with the standard deviation of at least three independent measurements.

confirming the rigidity of our catalyst (Supplementary Fig. 24). We also assessed cycling stability in GDE (Fig. 2f). Amazingly, $Pd_4Cu_1$-$Ni(OH)_2$ composite catalyst can stably sustain continuous 380 h test without obvious urea partial current density and urea yield rate decay. The service life of $Pd_4Cu_1$-$Ni(OH)_2$ catalyst is an order of magnitude higher than that of the reported catalysts (Supplementary Table 3, typically ≤30 h).

## Mechanistic study

Upon assessing the performance of urea electrosynthesis, it is essential to decode the unique role of copper single-atom alloy in C−N coupling. Considering the variety of by-products involved in $NO_3RR$ and $CO_2RR$ processes, FE is an important indicator to examine the influence of atomically dispersed Cu atoms in Pd host in urea electrosynthesis (Supplementary Figs. 25–28). Electrochemical performance of $Pd_4Cu_1$-$Ni(OH)_2$ sample in solo $NO_3RR$ or $CO_2RR$ was firstly assessed, $NH_3$ and CO were the main products (Supplementary Fig. 29), respectively. Notably, $NH_3$ and CO yield rates are much higher than urea yield rates,

suggesting C−N coupling toward urea synthesis possesses sluggish kinetics, consistent with LSV curves (Fig. 2a). The results also indicate that the co-reduction of $NO_3^-$ and $CO_2$ inhibits the single $NO_3RR$ or $CO_2RR$. Figure 3a–c show the FEs of the primary products for Pd-$Ni(OH)_2$, $Pd_1Cu_1$-$Ni(OH)_2$, $Pd_4Cu_1$-$Ni(OH)_2$ composite catalysts, respectively. $NO_2^-$ FEs are dominated between −0.1 and −0.6 V for Pd-$Ni(OH)_2$ sample, suggesting that metallic Pd catalyst enclosed by (111) plane can catalyze the conversion of $NO_3^-$ to $NO_2^-$, and the deep reduction of $NO_2^-$ to $NH_3$ process is interrupted (Fig. 3a)[27]. Notably, CO and urea synchronously emerge at −0.3 V, that is because $CO_2RR$ is triggered at more negative potential[28,29]. The result also indicates that the production of CO is a prerequisite for C−N coupling toward urea formation[14,17]. As shown in Fig. 3b, the formation of CO and urea is synchronously advanced to −0.2 V on $Pd_1Cu_1$-$Ni(OH)_2$ sample, further supporting the conclusion. In addition, $NH_3$ FEs all increase compared with that of Pd-$Ni(OH)_2$ between −0.1 and −0.6 V. That is because Cu is active for $NO_3RR$ to $NH_3$, and alloying Cu atoms in Pd lattice facilitates the deep reduction of $NO_2^-$ to $NH_3$[25]. Accordingly, urea FE increases

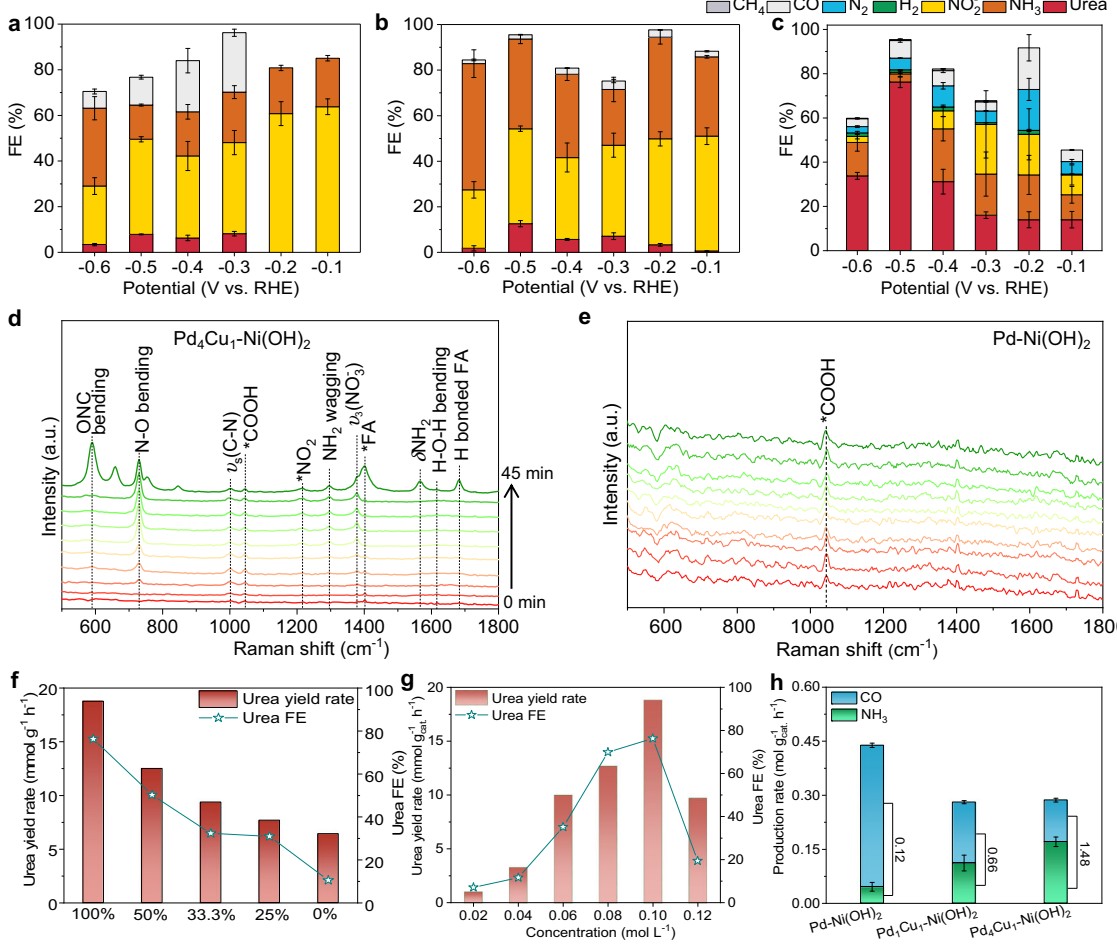

**Fig. 3 | Mechanistic study.** FEs of the primary products in urea electrosynthesis for **a** Pd-Ni(OH)$_2$, **b** Pd$_1$Cu$_1$-Ni(OH)$_2$, and **c** Pd$_4$Cu$_1$-Ni(OH)$_2$ composite catalysts assessed in 0.1 M KHCO$_3$ + 0.1 M KNO$_3$ (catalyst loading: 0.1 mg cm$^{-2}$). Time-resolved in situ Raman spectra recorded in urea electrosynthesis at −0.5 V from 0 to 45 min: **d** Pd$_4$Cu$_1$-Ni(OH)$_2$, **e** Pd-Ni(OH)$_2$. Urea yield rates and urea FEs **f** at different CO$_2$ partial pressure and **g** different concentrations of NO$_3^-$ for Pd$_4$Cu$_1$-Ni(OH)$_2$ at −0.5 V. **h** Production rates of CO and NH$_3$ in solo CO$_2$RR and NO$_3$RR, and the corresponding ratios of NH$_3$:CO at −0.5 V. **a**–**c**, **h** Error bars in accordance with the standard deviation of at least three independent measurements.

from 7.9% of Pd-Ni(OH)$_2$ to 12.6% of Pd$_1$Cu$_1$-Ni(OH)$_2$ at −0.5 V, verifying that the enhanced NO$_3$RR facilitates urea synthesis. It is reasonable to infer that the key N-intermediate for C−N coupling comes from the conversion process of NO$_2^-$ to NH$_3$, not NO$_2^-$. As Cu doping level in Pd lattice declines to Pd$_4$Cu$_1$, namely Cu single-atom alloy, urea FEs all greatly increase and the FEs of by-products (e.g., NO$_2^-$, NH$_3$, CO) decrease between −0.1 and −0.6 V (Fig. 3c). The optimal urea FE reaches 76.2% at −0.5 V, while NH$_3$ FE decreases to 3.7%. Moreover, a very small percentage of methane arises between −0.1 and −0.3 V for Pd$_4$Cu$_1$-Ni(OH)$_2$. From the above results, we can conclude that NO$_3$RR is greatly enhanced, and then C−N coupling toward urea formation is boosted.

To figure out the possible C- and N-intermediates for C−N coupling, a list of control experiments were carried out. As shown in Table 1, the possible C-intermediates, e.g., HCOOH, CH$_3$OH, HCHO and CO were employed as C-feeding, while NO$_3^-$ was employed as N-feeding. From entry 1-5, urea is obtained using HCOOH and CO as C-feeding. It is generally accepted that CO is the downstream reduction product of CO$_2$RR (CO$_2$ to *COOH to *CO)[30]. Therefore, we can conclude that *CO is the C-intermediate for C−N coupling toward urea synthesis, consistent with FEs result. Meanwhile, a series of N-intermediates, e.g., NO$_2^-$, NH$_2$OH, HCONH$_2$ (formamide, FA), NH$_3$, NH$_4^+$, were employed to replace NO$_3^-$. From entry 6–10, urea is only detected in the electrolytes with NO$_2^-$, NH$_2$OH or HCONH$_2$. Obviously, urea is

not formed by C−N coupling with NH$_3$ or NH$_4^+$ as N-intermediates. From entry 8, we infer that *CONH$_2$ may be the possible intermediate in urea synthesis, which is considered to be formed by a nucleophilic attack coupling of *CO and *NH$_2$[31]. As such, *NH$_2$ and *CO are N-intermediates and C-intermediates for C−N coupling toward urea formation.

To reveal C−N coupling mechanism on Pd$_4$Cu$_1$-Ni(OH)$_2$ sample, in situ Raman spectroscopic characterization was performed to trace the evolution of C- and N-species. Figure 3d, e and Supplementary Fig. 30 show the time-resolved Raman spectra in urea electrosynthesis at −0.5 V, recorded on Pd$_4$Cu$_1$-Ni(OH)$_2$, Pd-Ni(OH)$_2$ and Pd$_1$Cu$_1$-Ni(OH)$_2$, respectively. As shown in Fig. 3d, vibrational peaks located at 730 and 1378 cm$^{-1}$ can be attributed to a N−O bending mode and $\nu_3$ mode of free NO$_3^-$, respectively[32,33]. The intensity of the two peaks gradually increases with reaction time, suggesting the enrichment of NO$_3^-$ on catalyst surface[34]. Two vibrational peaks located at 1216 and 1296 cm$^{-1}$ synchronously appear at 20 min, which are assigned to *NO$_2$ and *NH$_2$ wagging modes, respectively[35,36]. It suggests that NO$_3^-$ is reduced to *NO$_2$, and then to *NH$_2$. A vibrational peak located at 1000 cm$^{-1}$ ascribing to $\nu_s$(C−N) mode of urea arises at 10 min, validating the formation of urea[37]. When the reaction proceeded to 45 min, vibrational peaks located at 590, 1402, 1567, 1683 cm$^{-1}$ appeared with high intensity, which can be attributed to OCN bending mode, C−H in-plane bending mode, $\delta$NH$_2$ of formamide (FA) and H bonded FA signal

**Table 1 | The list of control experiments carried out to elucidate the mechanistic pathway towards urea at −0.5 V for 2 h**

| Entry | C-source | N-source | Urea? | Electrolyte solution |
|---|---|---|---|---|
| 1 | $CO_2$ | $KNO_3$ | √ | 100 mM $KNO_3$ |
| 2 | HCOOH | $KNO_3$ | √ | 100 mM $KNO_3$ + 20 mM HCOOH |
| 3 | HCHO | $KNO_3$ | × | 100 mM $KNO_3$ + 20 mM HCHO |
| 4 | $CH_3OH$ | $KNO_3$ | × | 100 mM $KNO_3$ + 20 mM $CH_3OH$ |
| 5 | CO | $KNO_3$ | √ | 20 mM $KNO_3$ |
| 6 | $KHCO_3$ + $CO_2$ | $KNO_2$ | √ | 20 mM $KNO_2$ + 100 mM $KHCO_3$ |
| 7 | $KHCO_3$ + $CO_2$ | $NH_2OH$ | √ | 20 mM $NH_2OH$ + 100 mM $KHCO_3$ |
| 8 | $KHCO_3$ + $CO_2$ | $HCONH_2$ | √ | 20 mM $HCONH_2$ + 100 mM $KHCO_3$ |
| 9 | $KHCO_3$ + $CO_2$ | $NH_3$ | × | 20 mM $NH_3$ + 100 mM $KHCO_3$ |
| 10 | $KHCO_3$ + $CO_2$ | $NH_4Cl$ | × | 20 mM $NH_4Cl$ + 100 mM $KHCO_3$ |

(Supplementary Table 4), respectively[38]. The emergence of FA signal indicates that FA is really the intermediate product of C−N coupling toward urea formation. Notably, FA usually exhibits stronger Raman signal intensity than urea, which well explains the sudden emergence of a strong FA signal on $Pd_4Cu_1$-Ni(OH)$_2$ (Supplementary Fig. 31). Beyond that, a vibrational peak located at 1046 cm$^{-1}$ appears at 10 min, which is assigned to *COOH rooted from $CO_2$RR[39].

For $Pd_1Cu_1$-Ni(OH)$_2$ sample, the vibrational signals of *NO$_2$ and *NH$_2$ arise at 45 min with lower intensity, suggesting that the conversion of NO$_3^-$ to *NO$_2$ and *NO$_2$ to *NH$_2$ possess sluggish kinetics on $Pd_1Cu_1$ alloy (Supplementary Fig. 30). $\nu_s$(C−N) vibrational peak of urea can hardly be observed, suggesting that trace of urea is formed on $Pd_1Cu_1$ clusters. The characteristic vibrational peaks of FA, i.e., OCN bending mode, C–H in-plane bending mode, $\delta NH_2$ and H bonded FA, are also observed. The result indicates that the formation of urea on $Pd_1Cu_1$ alloy undergoes the similar pathway with Cu single-atom alloy. Furthermore, the signal of *COOH appears in the initial 5 min, indicating that $CO_2$ reduction to *COOH is not affected on $Pd_1Cu_1$ alloy. As such, the sluggish reduction kinetics of NO$_3^-$ to *NH$_2$ is the possible reason for the low urea yield on $Pd_1Cu_1$ alloy. As a stark contrast, only *COOH is observed for Pd-Ni(OH)$_2$, no *NO$_2$ and *NH$_2$ signal appear, suggesting NO$_3$RR is inhibited on metallic Pd (Fig. 3e), further verifying single-atom Cu in Pd lattice facilitates NO$_3$RR and then urea synthesis.

We further examined the evolution of Raman signal of *CO, which is the key C-intermediates for C−N coupling. As shown in Supplementary Fig. 32, the bridged *CO located at 2080 cm$^{-1}$ on $Pd_4Cu_1$-Ni(OH)$_2$ sample exhibits weaker Raman vibrational signal than $Pd_1Cu_1$-Ni(OH)$_2$ and Pd-Ni(OH)$_2$[40]. That is because the produced *CO is quickly consumed by *NH$_2$ for C−N coupling. For metallic Pd catalyst, two vibrational peaks located at 2050 and 2135 cm$^{-1}$ arose at 35 and 40 min, which were assigned to bridge type and linear type *CO, respectively[41]. From the above Raman spectroscopic results, we can conclude that *NO$_2$ to $NH_3$ in NO$_3$RR is inhibited on metallic Pd surface, which could not provide sufficient *NH$_2$ species for further C−N coupling. As such, CO and NO$_2^-$ are the primary products in the co-reduction of $CO_2$ and NO$_3^-$, well explaining high CO and low urea FEs on Pd-Ni(OH)$_2$ sample. When Cu is doped in Pd lattice to form $Pd_1Cu_1$ alloy, NO$_3$RR conversion is promoted and urea yield rate increases accordingly. As the Cu doping level is reduced to atomic dispersion, *NO$_2$ to $NH_3$ and C−N coupling processes are all accelerated, and urea yield rate and FE are boosted.

From the above results, we can infer the kinetics of $CO_2$RR and NO$_3$RR determines the final urea electrosynthesis. To confirm the conclusion, we further regulated the kinetics of $CO_2$RR and NO$_3$RR by changing $CO_2$ partial pressure or the concentration of NO$_3^-$ to slow down $CO_2$ and NO$_3$RR kinetics. As shown in Fig. 3f, g, urea yield rates and urea FEs all show decreasing trend with the $CO_2$ partial or NO$_3^-$ concentrations, suggesting that the kinetics of $CO_2$RR and NO$_3$RR

indeed determines urea electrosynthesis. Then, $NH_3$ and CO yield rates were obtained to investigate the impact of reduction kinetics (NO$_3$RR and $CO_2$RR) on urea electrosynthesis. As shown in Fig. 3h, $NH_3$ yield rates increase from 0.046 to 0.112 and 0.171 mol g$_{cat.}^{-1}$ h$^{-1}$, and CO decreases from 0.392 to 0.169 and 0.115 mol g$_{cat.}^{-1}$ h$^{-1}$ for Pd-Ni(OH)$_2$, $Pd_1Cu_1$-Ni(OH)$_2$ and $Pd_4Cu_1$-Ni(OH)$_2$ at −0.5 V, respectively. Surprisingly, the ratio of $NH_3$:CO yield rates for $Pd_4Cu_1$-Ni(OH)$_2$ is 1.5, approaching the theoretical value of 2 in urea. The result clarifies the matched kinetics of NO$_3$RR and $CO_2$RR contributes the high urea yield rate and FE in C−N coupling process.

## Theoretical calculations

Then, density functional theory calculations were carried out to reveal the promotion effect of Cu single-atom alloy on urea electrosynthesis. According to the HRTEM result, single-atom Cu alloyed Pd(111) (denoted as $Cu_1Pd$) and Pd(111) planes were employed as the slabs. Differential charge density plots of $Cu_1Pd$(111) (Fig. 4a) indicate that the electrons of Cu are delocalized and donated to Pd atoms around Cu atom due to higher electronegativity of Pd atoms[42]. Bader charge analysis confirms Cu atom denotes 0.21 e$^-$ to adjacent Pd atoms on $Cu_1Pd$(111) plane, while Pd(111) plane still shows balanced electron distribution (Supplementary Fig. 33). Given C- and N-intermediates for C−N coupling, *NH$_2$ is nucleophilic and *CO is electrophilic. Therefore, *NH$_2$ prefers to adsorb on Cu sites while *CO on Pd sites. To confirm this conclusion, differential charge density plots of Pd(111)-*NH$_2$, Pd(111)-*CO, $Cu_1Pd$(111)-*NH$_2$ and $Cu_1Pd$-*CO were obtained (Fig. 4b). The results indicate that *NH$_2$ bonded to Pd-Cu atoms exhibits larger electron transfer, indicating strong tendency to bond. The adsorption energy also supports this conclusion (*NH$_2$ on Cu: −2.59 eV, *CO on Cu: −2.16 eV). Similarly, *CO tends to adsorb on adjacent two Pd atoms (Supplementary Figs. 34−36).

To further understand the promotion effect of Cu single-atom alloy on urea electrosynthesis, we firstly derived the free-energy diagram ($\Delta G$) of reaction profile for each elementary step in $CO_2$RR. As shown in Fig. 4c, $CO_2$ adsorption on the catalyst surface and desorption of *CO are two endothermic processes, the later possesses larger energy barrier which is potential-determining step (PDS) for $CO_2$RR to CO (Supplementary Table 5). $Cu_1Pd$(111) plane lowers energy barrier of $CO_2$ adsorption process and lifts the $\Delta G$ of *CO desorption process. It means that Cu single-atom alloy facilitates the conversion of $CO_2$ to *CO, but restrains *CO desorption from catalyst surface. As such, C−N coupling is promoted and CO FE is declined. Then, the free-energy diagram in electrochemical NO$_3$RR was also obtained, in which *NO$_2$ was selected as the initial species (Fig. 4d and Supplementary Table 6). *NO$_2$ → *NO$_2$H, *NO → *HNO and *NH$_3$ → * + $NH_3$ processes are endothermic processes. *NO → *HNO process exhibits the largest energy barrier, which is PDS step in NO$_3$RR. The energy barrier is 0.74 eV on $Cu_1Pd$(111) surface, much lower than that on Pd(111) surface (1.15 eV), which accounts for the preference for *NH$_2$ formation on Cu single-atom alloy. The first C−N coupling process of *NH$_2$ + *CO → *CONH$_2$ is typically endothermic reaction. And the second C−N coupling process is exothermic reaction with large energy output up to 6.44 eV on $Cu_1Pd$(111) surface. The energy barriers are 0.07 and 0.19 eV on $Cu_1Pd$(111) and Pd(111) surface, respectively, which validates Cu single-atom alloy facilitates C−N coupling. The most stable adsorption configurations on $Cu_1Pd$(111) and Pd(111) planes are demonstrated in Fig. 4e and Supplementary Fig. 37. Although Cu(111) planes deliver much lower energy barrier of PDS (0.45 eV), $\Delta G$ of the first C−N coupling step on Cu(111) planes is the largest, which leads to negligible urea formation on Cu nanosheets (Supplementary Fig. 38).

## Promoting urea electrosynthesis performance by optimizing the carrier

Upon clarifying the promotion effect of Cu single-atom alloy on urea electrosynthesis, we further uncovered the role of Ni(OH)$_2$ carrier on

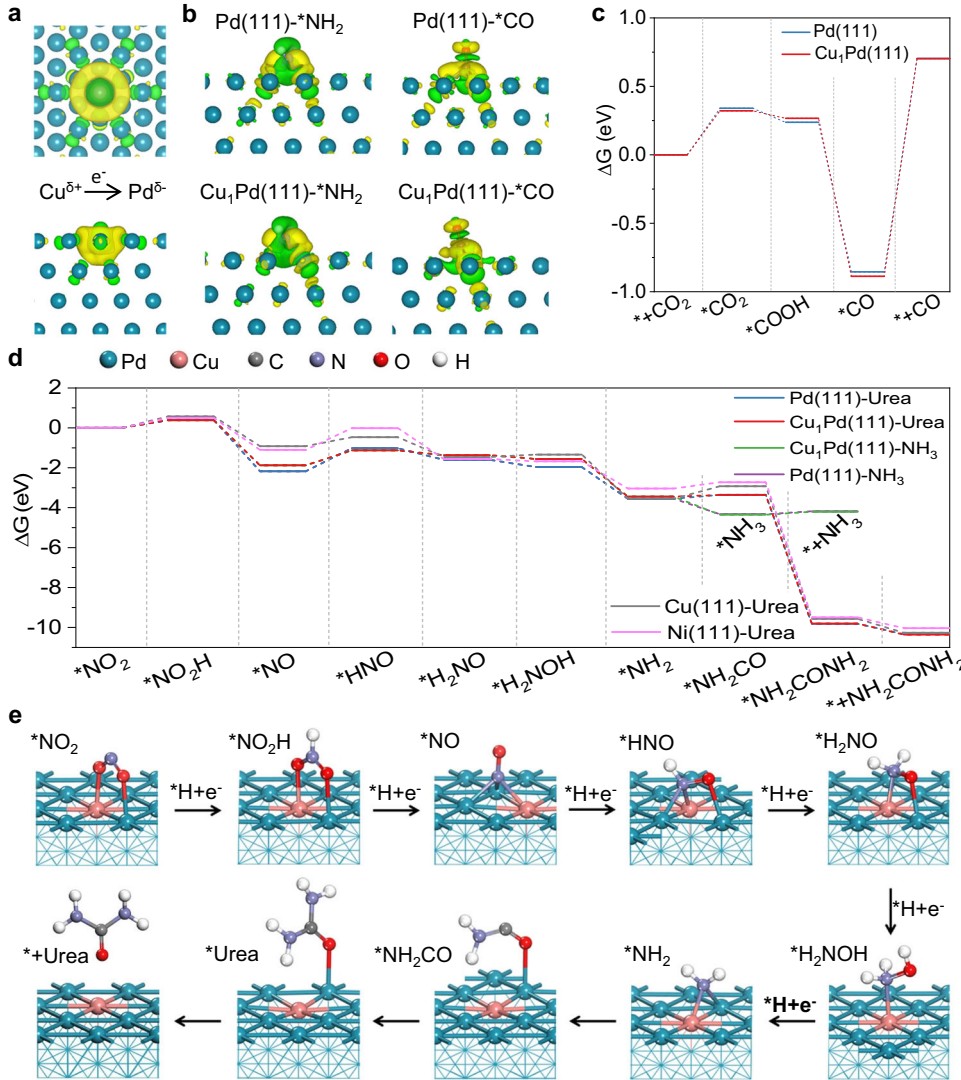

**Fig. 4 | Theoretical calculations. a** Differential charge density of Cu₁Pd(111) (top view: top, side view: down). The isosurface value of yellow contour is 0.001 e/bohr³. **b** Differential charge density of Cu₁Pd(111)-*NH₂, Pd(111)-*NH₂, Cu₁Pd(111)-*CO, Pd(111)-*CO. The isosurface values of yellow contour are 0.002 or 0.00157 e/bohr³, respectively. **c** Energy profiles of each elementary step in single CO₂RR catalyzed by Cu₁Pd(111) and Pd(111) planes. **d** Energy profiles of each elementary step in NO₃RR with C−N coupling toward urea synthesis catalyzed by Cu₁Pd(111), Pd(111), Cu(111) and Ni(111) planes. **e** DFT-calculated urea synthesis cycle on Cu₁Pd(111) surface.

urea electrosynthesis. First, Pd₄Cu₁ anchored on Ni(OH)₂ nanosheets suppress the aggregation of clusters during long-term electrochemical process, which contributes to the good cycling stability. Second, Pd₄Cu₁/Ni(OH)₂ interface facilitates the dissociation of interfacial water molecules by forming Ni^{δ+}···O^{2−}H···Pd₄Cu₁ interaction in alkaline electrolyte (Supplementary Fig. 39)[43,44]. As such, more active H atoms are formed on Pd₄Cu₁ catalyst surface, and then the following deoxyreduction processes (CO₂ → *CO, NO₃⁻ → *NH₂) in urea formation are accelerated. This conclusion is confirmed by replacing Ni(OH)₂ nanosheets with good conductors (reduced graphene oxide, rGO and XC-72) or semiconductor (TiO₂ nanosheets) as carriers (Supplementary Figs. 40 and 41). Given the important promotion role of Ni(OH)₂ carrier in water splitting, we infer that urea yield rate can be further improved by Fe³⁺ doping in Ni(OH)₂ nanosheets, as high valence state of Fe³⁺ in Ni(OH)₂ was proved to improve water splitting[45]. Theoretical calculation results reveal that water molecules indeed tend to adsorb on Ni(OH)₂ or Fe-doped Ni(OH)₂ surface by forming Ni−OH₂ or Fe−OH₂ interaction (Fig. 5a, b). As such, the energy barrier for breaking H−OH bond declines from 0.27 eV on Cu₁Pd surface to −0.25 and −0.27 eV on Cu₁Pd/Ni(OH)₂ and Cu₁Pd/FeNi(OH)₂ interface (Fig. 5c), respectively, suggesting that water splitting is boosted on the interface. Notably, the

produced active H atoms on Cu₁Pd surface tend to combine with the adjacent *NO₃ and *CO₂, instead of coupling each other to release H₂, which well explains the high urea FE for P₄Cu₁-Ni(OH)₂ (Supplementary Figs. 42 and 43). Hence, Pd₄Cu₁ single-atom alloy clusters anchored on Fe-doped Ni(OH)₂ composite sample was synthesized, denoted as Pd₄Cu₁-FeNi(OH)₂ (Fig. 5d and Supplementary Figs. 44–47). The control experiments confirm that Fe-doped Ni(OH)₂ nanosheets carriers are inert for CO₂RR and have weak ability to catalyze NO₃RR and urea formation, further verifying Pd₄Cu₁ clusters are the real active sites for C−N coupling (Supplementary Figs. 48–50)[46]. To confirm the enhanced water dissociation speeds up urea formation, D₂O was employed as D-source which can slow down D-OD dissociation and D transfer processes due to isotope effect[47]. As shown in Fig. 5e, urea yield rate and urea FE are declined to 1/6 with D₂O as D-source. As such, the kinetics of CO₂RR and NO₃RR are enhanced after Fe³⁺ doping in Ni(OH)₂ nanosheets, which is validated by both improved NH₃ and CO yield rates (Fig. 5f).

As expected, urea yield rate reaches 63.5 mmol g$_{cat.}$⁻¹ h⁻¹ with FE of 59.7% in H-type cell at −0.6 V (V vs. RHE), it is approximately 3.4-fold larger than that of Pd₄Cu₁-Ni(OH)₂ recorded at −0.5 V (Supplementary Fig. 51). To further maximize energy utilization efficiency, urea

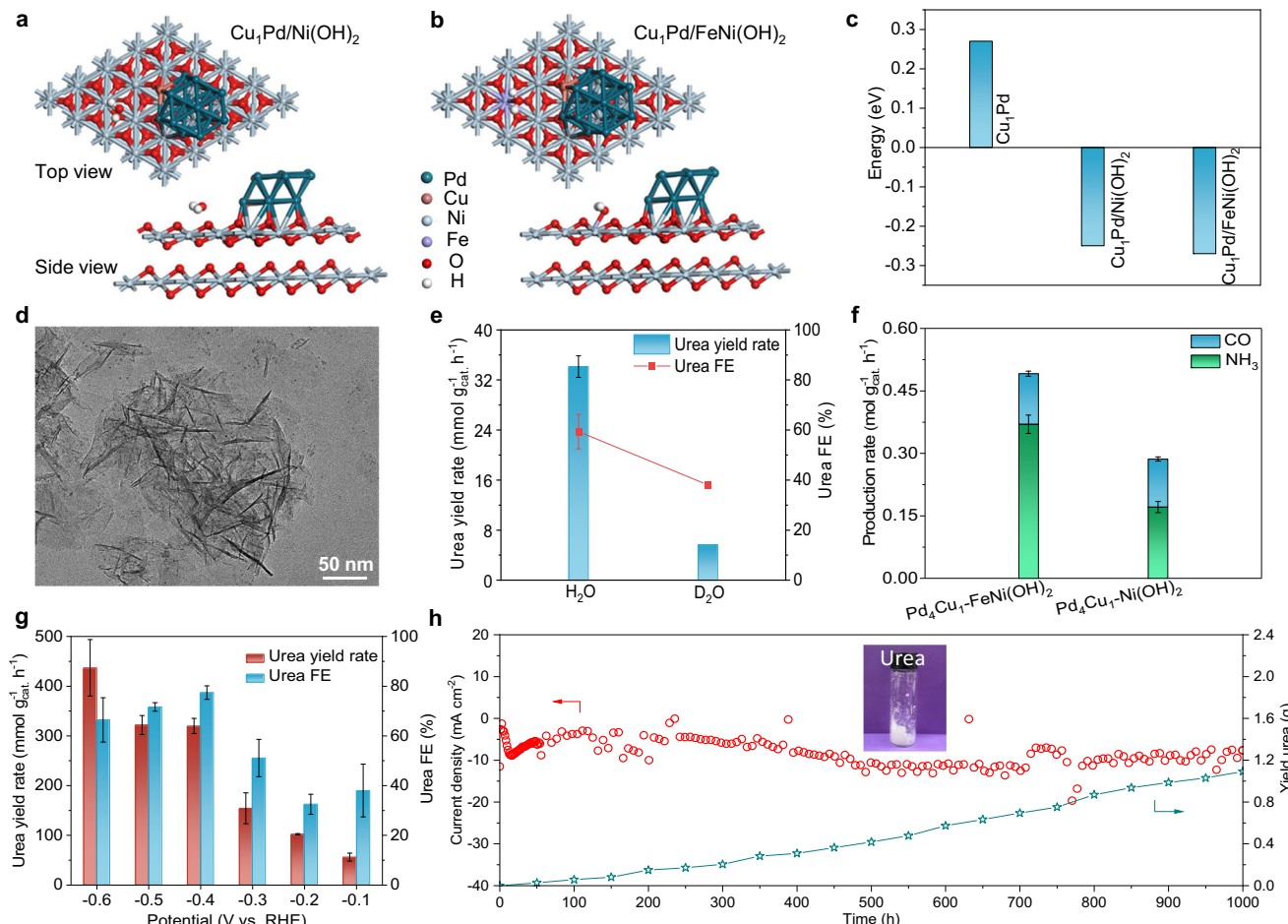

**Fig. 5 | Characterization of Pd4Cu1-FeNi(OH)2 sample.** Adsorption configurations of $H_2O$ on **a** $Cu_1Pd/Ni(OH)_2$ and **b** $Cu_1Pd/FeNi(OH)_2$ interface. **c** The energy barrier of dissociation of H−OH bond on $Cu_1Pd$ surface, $Cu_1Pd/Ni(OH)_2$ and $Cu_1Pd/FeNi(OH)_2$ interfaces. **d** TEM image of $Pd_4Cu_1$-FeNi(OH)₂ sample. **e** Urea yield rates and urea FEs with $H_2O$ or $D_2O$ as H-source. **f** The comparison of production rates of CO and $NH_3$ with $Pd_4Cu_1$-FeNi(OH)₂ or $Pd_4Cu_1$-Ni(OH)₂. **g** Potential-dependent urea yield rates and FEs assessed in GDE coupled with oxidation of anisyl alcohol at anode, **h** long-term $I$-$t$ stability test and the time-resolved urea yield amount for $Pd_4Cu_1$-FeNi(OH)₂ at −0.5 V in the mixture of 0.1 M KHCO₃ + 0.1 M KNO₃ using a continuous flow system in GDE with $CO_2$ bubbling (20 mL min⁻¹) and catalyst loading of 0.025 mg cm⁻². Insert: the produced urea. **e**–**g** Error bars in accordance with the standard deviation of at least three independent measurements.

electrosynthesis in GDE was also assessed by coupling the oxidation of anisyl alcohol at anode (Supplementary Fig. S52)[48]. As shown in Fig. 5g, the best urea yield rate and FE reach recorded 436.9 mmol $g_{cat.}^{-1}$ h⁻¹ and 66.5% at −0.6 V, it is about an order of magnitude higher than the optimal urea yield rate that has been reported (Supplementary Table 3). Beyond that, $Pd_4Cu_1$-FeNi(OH)₂ composite catalyst delivers astounding cycling stability, which can sustain continuous 1000 h test without obvious current decay (Fig. 5h). The produced amount of urea in the electrolyte is proportional to the reaction time, further confirming the rigidity of our composite catalyst. Finally, 1.05 g urea was obtained from the electrolyte (Supplementary Fig. S53).

## Discussion

In summary, highly efficient Cu single-atom alloy catalyst is synthesized for urea electrosynthesis with $CO_2$ and $NO_3^-$ from dynamics and thermodynamics points. In situ Raman spectroscopic results reveal the key coupling pathway of $*CO + *NH_2 → *NH_2CO + *NH_2 → NH_2CONH_2$. Theoretical calculation results indicate that Cu single-atom alloy in Pd lattice facilitates the further reduction of $NO_2^-$ to $NH_3$ and lowers the energy barrier for the first C−N coupling. In addition, Cu doping level and the interface of $Pd_4Cu_1/FeNi(OH)_2$ tunes the kinetics of $CO_2RR$ and $NO_3RR$ to achieve the matched formation kinetics of $*CO$ and $*NH_2$. Taken together, $Pd_4Cu_1$-FeNi(OH)₂ composite catalyst achieve a high urea yield rate of 436.9 mmol $g_{cat.}^{-1}$ h⁻¹ and 66.5% in GDE, as well as long

cycling stability of 1000 h, far exceeding the reported results. This work provides an insight into catalyst design toward highly efficient, selective and robust C−N coupling from the angle of single-atom alloy.

## Methods
### Synthesis of α-Ni(OH)₂ nanosheets
Ni(NO₃)₂·6H₂O (1.45 g) and urea (0.6 g) were firstly dissolved in a mixture of triethylene glycol (40 mL) and DI water (10 mL) to form a light green transparent solution. Then, the solution was transferred and sealed in an autoclave with a Teflon liner and was heated at 120 °C for 24 h. After it was cooled to room temperature, the product was collected by centrifugation and further soaked in ethanol for 24 h. Finally, the product was collected by centrifugation and washed with ethanol for three times, dried in a vacuum oven for 24 h.

### Synthesis of Fe-doped Ni(OH)₂ nanosheets
Ni(NO₃)₂·6H₂O (1.45 g), urea (0.6 g) and FeCl₃·6H₂O (405.5 mg) were firstly dissolved in a mixture of triethylene glycol (40 mL) and water (10 mL) to form a light yellow transparent solution. Then, the solution was transferred and sealed in an autoclave with a Teflon liner, and was heated at 120 °C for 24 h. After it was cooled to room temperature, the product was collected by centrifugation and further soaked in ethanol for 24 h. Finally, the product was collected by centrifugation and washed with ethanol for three times, and dried in a vacuum oven for 24 h.

## Synthesis of $Pd_xCu_1$-$Ni(OH)_2$ composite catalysts

In a typical synthesis of $Pd_4Cu_1$-$Ni(OH)_2$ composite structure, $Ni(OH)_2$ (35.3 mg) nanosheets powder was ultrasonically dispersed in 20 mL DI water for 5 min. Then, $K_2PdCl_4$ (3.13 mg) and $CuCl_2 \cdot 2H_2O$ (0.4 mg) were dissolved in the above mixture solution. After that, ice water cooled $NaBH_4$ solution (1.0 mM, 6 mL) was dropped in the mixture to reduce $Pd^{2+}$ and $Cu^{2+}$ to form $Pd_4Cu_1$ alloy cluster. After stirring for another 1 h, the final product was collected by centrifugation, washed three times with ethanol and water, and dried in a vacuum oven for 24 h. The protocol for the synthesis of $Pd_xCu_1$-$Ni(OH)_2$ (x = 1, 2, 3, 5, 6) was similar with that of $Pd_4Cu_1$-$Ni(OH)_2$ except with Cu and Pd dosage of 1.0, 0.7, 0.5, 0.34, 0.3 mg and 2.0 2.6, 2.9, 3.3, 3.4 mg, respectively. The protocols for the synthesis of $Pd_4Cu_1$-XC-72 and $Pd_4Cu_1$-$TiO_2$ were similar with that of $Pd_4Cu_1$-$Ni(OH)_2$ except with XC-72 (35.3 mg) and $TiO_2$ nanosheets (35.3 mg) as carriers, respectively. The protocols for the synthesis of $Pd_4Cu_1$-$FeNi(OH)_2$ was similar except with $FeNi(OH)_2$ nanosheets (35.3 mg) as carrier and $NaBH_4$ solution (1.0 mM, 18 mL).

## Electrosynthesis of urea in H-type cell

$Pd_4Cu_1$-$Ni(OH)_2$ (2 mg) was ultrasonically dispersed for 30 min in a mixture of $H_2O$ (0.7 mL), isopropanol (0.25 mL) and Nafion (0.05 mL, 5 wt.%) to form the catalyst ink. Then, a 50-μL aliquot of catalyst ink was coated evenly on carbon paper with an area of $1 \times 1$ $cm^2$ (catalyst loading: 0.1 mg $cm^{-2}$) and dried under infrared lamp, which was used as working electrode. An Ag/AgCl and Pt plate were used as reference electrode and counter electrode, respectively.

All electrochemical tests were performed in an H-type cell using three-electrode system at room temperature, in which cathode chamber and anode chamber were separated by a commercial Nafion 117 membrane. The electrolyte solution for both cathode and anode was the mixture of $KHCO_3$ (40 mL, 0.1 M) and $KNO_3$ (0.1 M) solution. Prior to the electrochemical test, electrolyte was bubbled with continuous ultra-high purity $CO_2$ gas (99.999%) for 30 min. Electrochemical coupling of $CO_2$ and $NO_3^-$ was triggered under constant potentials (−0.1, −0.2, −0.3, −0.4, −0.5, and −0.6 V, versus the reversible hydrogen electrode, RHE) with continuous $CO_2$ flow. The applied potentials were all converted to the RHE scale according to the following equation:

$$E(vs.RHE) = E(vs.Ag/AgCl) + 0.197\,V + 0.059\,V \times pH \tag{1}$$

After 2 h of continuous electrolysis, the produced urea in the electrolyte at the cathode chamber was spectrophotometrically quantified with diacetylmonoxime reagent or determined by hydrogen nuclear magnetic resonance ($^1$H-NMR) spectroscopy measurement. The possible liquid byproducts, e.g., $NH_3$, $NO_2^-$, in the electrolyte were spectrophotometrically quantified with Nessler reagent, indophenol blue and Griess reagent, respectively. The possible gaseous byproducts, e.g., CO, $CH_4$, $H_2$ and $N_2$, were quantified by gas chromatography (GC). The electrochemical performance for other electrocatalysts were also assessed using the similar method with $Pd_4Cu_1$-$Ni(OH)_2$.

## Electrosynthesis of urea in GDE

Electrosynthesis of urea in a flow cell was performed to improve carbon dioxide mass transfer kinetics. Anisyl alcohol oxidation was coupled at anode in flow cell to further reduce overpotential. To prepare catalyst ink, $Pd_4Cu_1$-$Ni(OH)_2$ (2 mg) was dispersed in a mixture of Nafion (5 wt.%, 50 μL), isopropanol (250 μL) and $H_2O$ (700 μL), and ultrasound for 30 min. Then the catalyst ink (50 μL) was uniformly coated on the hydrophobic carbon paper with an area of $1 \times 1$ $cm^2$ and catalyst loading of 0.1 mg $cm^{-2}$, which was used as working electrode. An Ag/AgCl and Pt flake were used as reference electrode and counter electrode, respectively. The electrolyte solution used at cathode was the mixture of $KNO_3$ (40 mL, 0.1 M) and $KHCO_3$ (0.1 M). The electrolyte

solution used at anode was the mixture of KOH (50 mL, 0.1 M) and anisyl alcohol (2.5 mL). The flow rates of the electrolyte solution were all 60 mL $min^{-1}$ both for anode and cathode, and $CO_2$ was continuous pumped with a flow rate of 20 mL $min^{-1}$. The volume of the cathode and anode chambers was $1 \times 1 \times 1$ $cm^3$. When the flow cell was successfully assembled and stably operated, electrosynthesis of urea was triggered by applying a fixed potential versus RHE at cathode. After 1 h of continuous electrolysis, the produced urea in the electrolyte was spectrophotometrically quantified with diacetylmonoxime reagent or determined by $^1$H-NMR spectroscopy measurement. The procedure for $Pd_4Cu_1$-$FeNi(OH)_2$ was similar with that of $Pd_4Cu_1$-$Ni(OH)_2$ sample, except with the catalyst loading of 0.025 mg $cm^{-2}$.

## Determination of urea

**Way 1.** EDTA (0.1 g) was dissolved in urease solution (10 mL, 5 mg $mL^{-1}$). Then, electrolyte solution (1.8 mL) was added into the above solution (0.2 mL). The final solution was reacted for 40 min at 37 °C in a shaker. The produced $NH_3$ was spectrophotometrically quantified with indophenol blue method.

**Way 2.** The produced amount of urea in the electrolyte was determined by diacetylmonoxime method. Typically, 1 mL electrolyte was added into 2 mL acid-ferric solution (100 mL concentrated phosphoric acid, 300 mL concentrated sulfuric acid, 600 mL deionized water and 100 mg ferric chloride). And then 1 mL diacetylmonoxime (DAMO)-thiosemicarbazide (TSC) solution (5 g DAMO and 100 mg TSC were dissolved in 1000 mL deionized water) was added into the mixture. After that, the solution was heated to 100 °C and maintained for 20 min. After it was cooled to room temperature, UV–Vis absorption spectrum was performed and the absorbance at 525 nm was acquired. A series of standard urea solutions were used to obtain working curves for urea determination.

## Determination of ammonia ($NH_3$)

**Way 1.** The produced ammonia in the electrolyte solution was spectrophotometrically quantified with Nessler reagent. Typically, the diluted electrolyte solution (5 mL) was added into seignette salt solution (100 μL, 0.2 M) to wipe off the possible metal cations contamination. Commercial Nessler reagent (150 μL) was added into the above mixture for 10 min. Absorbance at 420 nm was acquired from the UV-Vis absorption spectrum. A series of standard $NH_3$ solutions were used to obtain working curve for $NH_3$ determination.

**Way 2.** Sodium salicylate (5 g) and seignette salt (5 g) were dissolved in NaOH solution (100 mL, 1 M) to obtain solution A. NaClO (3.5 mL, 10−15%) was diluted in 96.5 mL DI water to obtain solution B. Sodium nitroferricyanide (0.2 g) was dissolved in 20 mL DI water to obtain solution C. To quantify $NH_3$, solution A, solution B and solution C were added in turn in the diluted electrolyte solution (2 mL). After 2 h in a dark room at room temperature, absorbance at 662 nm was acquired from the UV-vis absorption spectrum. A series of standard $NH_3$ solutions were used to obtain working curve for $NH_3$ determination.

## Determination of nitrite ions ($NO_2^-$)

Nitrite ions were spectrophotometrically quantified with Griess reagent. Typically, Griess reagent (200 μL) was added into electrolyte solution (5 mL). Then, the solution was heated to 100 °C and maintained for 1 min. After it was cooled to room temperature, UV-Vis absorption spectrum was acquired and the absorbance at 540 nm was obtained. A series of standard $NO_2^-$ solutions were used to obtain working curve for $NO_2^-$ determination.

## Determination of $N_2$, $H_2$, CO and $CH_4$

The amounts of $N_2$, $H_2$, CO and $CH_4$ were quantified by gas chromatograph (GC) equipped with TCD and FID detectors.

The FEs for urea, $NO_2^-$, $NH_3$, $N_2$, CO, $CH_4$, and $H_2$ were calculated according to Eqs. (2)–(7):

$$FE_{urea} = (16F \times C_{urea} \times V)/(60.06 \times Q) \tag{2}$$

$$FE_{NO_2^-} = (2F \times C_{NO_2^-} \times V)/(47 \times Q) \tag{3}$$

$$FE_{NH_3} = (8F \times C_{NH_3} \times V)/(17 \times Q) \tag{4}$$

$$FE_{N_2} = (10F \times C_{N_2} \times V/V_m)/(28 \times Q) \tag{5}$$

$$FE_{CO} = (2F \times C_{CO} \times V/V_m)/(28 \times Q) \tag{6}$$

$$FE_{CH_4} = (8F \times C_{CH_4} \times V/V_m)/(16 \times Q) \tag{7}$$

Where $F$ is the Faraday constant (96485.3 C mol$^{-1}$) and $Q$ is the total charge passed through the working electrode.

$CO_2$-to-urea selectivity and $NO_3^-$-to-urea selectivities were calculated according to Eqs. 8 and 9:

$$N_{urea} - selectivity = n_{urea}(N)/n_{total}(N) \tag{8}$$

$$C_{urea} - selectivity = n_{urea}(C)/n_{total}(C) \tag{9}$$

### Theoretical calculation

The calculations in this work were performed with the Vienna ab initio Simulation Package (VASP), calculating the exchange-correlation function via the generalized gradient approximation (GGA) within the Perdew–Burke–Ernzerhof (PBE) flavor[49,50]. The Projected Augmented Wave (PAW) method was employed to describe the core-valance electron interaction[51,52]. The kinetic energy cutoff of 400 eV for plane-wave basis was set, and the reciprocal space was sampled by a $3 \times 3 \times 1$ Monkhorst–Pack grid of size. The $4 \times 4$ Pd(111) surface slabs were constructed with four layers (bottom two layers fixed), with vacuum layers of at least 15 Å to avoid the vertical interactions. The convergence criteria are $10^{-5}$ eV and 0.05 eV/Å for energy differences and atomic remaining force, respectively.

The binding energy is defined as $E_{Binding} = E_{A@Sub} - E_{Sub} - E_A$, where $E_{A@Sub}$ is the total energy of an $A$ intermediate adsorbed over the substrate, $E_{Sub}$ and $E_A$ are the entire energy of one single $A$ adsorbate and substrate in vacuum. The computational hydrogen electrode (CHE) model was applied for the simulation of the proton-coupled electron (H$^+$+e$^-$) transfer process via simplified the proton-coupled electron-transfer step to (H$^+$+e$^-$→1/2H$_2$). DFT calculated free energies ($G$) were corrected according to $G = E_{DFT} + E_{ZPE} - TS$ (298.15 K), where $E_{DFT}$ is the calculated total energy for each step, $E_{ZPE}$ is the zero-point energy and $S$ is the entropic contribution.

### Sample characterizations

Prior to electron microscopy characterizations, a drop of the suspension of nanostructures in ethanol was placed on a piece of carbon-coated copper grid and dried under ambient conditions. Transmission electron microscopy (TEM), high-resolution TEM (HRTEM) images and the corresponding energy-dispersive X-ray spectroscopy (EDS) mapping profiles were taken on a JEOL JEM-2100F field-emission high-resolution transmission electron microscope operated at 200 kV. Powder X-ray diffraction (XRD) patterns were recorded on a Philips X'Pert Pro Super X-ray diffractometer with Cu-Kα radiation ($\lambda = 1.5418$ Å). X-ray photoelectron spectra (XPS) were collected on an ESCALab 250 X-ray photoelectron spectrometer with nonmonochromatized Al-Kα X-ray as the excitation source. The concentrations of Pd and Cu were measured with a Thermo Scientific PlasmaQuad 3 inductively coupled plasma mass spectrometry (ICP-MS) after dissolving the samples with a mixture of HCl and $HNO_3$ (3:1, volume ratio). In situ Raman spectroscopy was performed with the Raman microscopy system (WITEC alpha300 R confocal Raman system) using 633 nm He–Ne laser as the excitation source.

## Data availability

The authors declare that all data supporting the findings of this study are available in the article and its Supplementary Information.

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

## Acknowledgements

This work was financially supported in part by the National Natural Science Foundation of China (22005268, 22206042), University Leading Talents Program of Zhejiang province (4095C502222140203, 4095C502222140201), Zhejiang Provincial Natural Science Foundation of China (LQ22B060007), and Medical Health Science and Technology Project of Zhejiang Provincial Health Commission (2023KY1009). We thank Dr. Liang Chen from Taiyuan University of Technology for the discussion of theoretical calculation.

## Author contributions

W.Y. and P.G. conceived the idea for this work. M.X. and F.W. prepared the catalysts and performed the characterizations and catalytic measurements. M.X. and Y.Z. performed in situ Raman experiments. L.C. carried out the DFT calculations. X.W., G.J., L.C. and Y.H. analyzed the results. Y.Y., G.Z., and X.L. participated in material characterization. M.X., P.G. and W.Y. wrote the manuscript. All the authors contributed to the interpretation of the data and preparation of the manuscript.

## Competing interests

The authors declare no competing interests.
