## [Peer Review File · Nature Communications]

Kinetically matched C–N coupling toward efficient urea electrosynthesis enabled on copper single-atom alloyREVIEWER COMMENTS

Reviewer #1 (Remarks to the Author):

The paper "Kinetically matched C–N coupling toward efficient urea electrosynthesis enabled on copper single-atom alloy" represents a high-efficiency urea production rare and high Faradic efficiency. The catalyst Pd₄Cu₁-FeNi(OH)₂ is well characterized and the authors achieved a high catalytic activity and stability in urea electrosynthesis. Despite this appreciable effort, some statements which are not really supported by solid proofs.

1. Isolated Pd or Cu is avoided during the co-reduction of Pd²⁺ and Cu²⁺ at various molar ratio, but exclusive PdCu single atom alloy. Please clarify the synthetic process.
2. From Fig. 1c, the distribution of Cu and Pd does not match well. It seems this result can not support the formation of PdCu single atom alloy.
3. From the R-space and WT counter plot, the spectra of Pd₄Cu is quite consistent with Cu foil. Generally, Pd should induce strain in Cu lattice, resulting the expansion in scattering path of Cu. See Nano Energy, 2022, 102, 107704; and Chem Catalysis 2021, 1, 1088–1103. Besides, The XAFS spectra of Pd is missing.
4. All the characterization, including in-situ Raman, only indicates the Pd₄Cu₁-Ni(OH)₂ is a superior urea-electrosynthesis catalyst, which is far from the topic of kinetic match C–N coupling.
5. The authors state that the introduction of Fe to forming Pd₄Cu-FeNi(OH)₂ could further improving the catalytic performance by 7-fold in urea production. The authors should affirm that whether the Pd₄Cu is the active site for urea electrosynthesis. The authors did not uncover the catalytic mechanism of the urea production over Pd₄Cu-Ni(OH)₂ or Pd₄Cu-FeNi(OH)₂.
6. Some expression are wrong, e.g. Na(OH)₂

For these reasons, I cannot recommend the paper in this form for publication in Nat. Commun.

Reviewer #2 (Remarks to the Author):

The authors report a single-atom copper-alloyed Pd catalyst (Pd₄Cu₁) that can achieve highly efficient C–N coupling toward urea electrosynthesis. The synthesized Pd₄Cu₁-FeNi(OH)₂ composite catalyst achieves a recorded urea yield rate of 436.9 mmol g⁻¹ h⁻¹ and FE of 66.4%, as well as ultralong cycling stability of 1000 h. There are several issues that need to be addressed and hence I recommend a major revision before being considered for publication.

Comments

1. What was the rationale behind going to single atom copper catalyst?
2. Were the experimental results compared with planar Cu, planar Ni and planar Pd? If not, please perform the needed experiments and show the trends of urea selectivity, FE and current density.
3. In line 71, please mention the total loading on the GDE.
4. Fig 1a shows the TEM image which is too unclear to analyze and conclude the uniform distribution. Moreover, they seem to be more like clusters instead of single atoms dispersed on the lattice. Please refer to this article as a reference to show TEM images for single atoms dispersion. Link: <https://www.frontiersin.org/articles/10.3389/fchem.2021.717201/full>
5. Fig 1b is not clear at all and authors have pointed to a measurement and referred to it as diameter, whereas nothing spherical is seen in this image. Please specify how you measured that distance d and what it denotes. Moreover, the image quality is very poor to analyze and conclude anything.

6. What does Fig 1c explain? Please explain the first and fourth part of figure 1c properly.
7. In line 112, authors mention that XPS and ICP-MS show the same results. Why were they both performed to validate the same results? What other insights did you get from each of them besides this?
8. What insights did wave transform analysis give for this study? The contours in Fig 1h don't explain or signify anything. Please put a clearer figure and explain what it signifies.
9. Perform the EXAFS for Pd foil and Ni foil as well to give a clear comparison (Fig 1f)
10. In fig 2b, show urea current density as well.
11. In Fig 2c and 2d provide the urea current density data as well.
12. In Fig 2e, why hasn't the urea yield rate been shown till 100 h. This is a multiple-product reaction unlike HER, hence stable total current density for 100h doesn't necessarily mean stable urea current for 100 hours. Also show urea current density along with yield rate for 100h to comment on the stability.
13. In figure 2f, show the urea current density and yield rate for 360 hours to comment on the stability of the GDE.
14. In figures 3a, b, c, please mention the current density of each of the products.
15. Perform the theoretical calculations for pure CU and NI as well to get a thorough comparison.
16. Was XAFS just done to confirm the fine structure of Pd? It looks like it was already established using other characterizations performed.
17. In line 167, authors conclude that alloying Cu single atoms in Pd lattice boosts urea electrosynthesis performance. Was dispersion of Pd atoms in Cu lattice tried to thoroughly conclude that the former arrangement performs better?
18. In line 171, authors mention how the urea FE and yield rates vary with potential. Please explain the reason for observed values as they don't seem to fall in a particular trend.
19. Metals like Fe, Co, Ni, Cu, Zn were chosen to form single metal alloys. Please explain the rationale behind choosing these metals.
20. FE of 69% at -0.4V vs RHE has already been achieved with Pd Cu bimetallic catalysts. Please refer <https://pubs.rsc.org/en/content/articlelanding/2023/ey/d2ey00038e> and explain what new insights does your work offer?
21. In the literature review table, please include the urea current densities as well.

Minor Comments

1. Line 10 has some errors which makes it hard to understand.
2. The spelling of conclusion is wrong in line 412.
3. The figure quality needs improvement.

Reviewer #3 (Remarks to the Author):

In this manuscript, Xu et al. studied the urea electrosynthesis from CO₂ and NO₃⁻ by using Pd₄Cu₁-Ni(OH)₂ or Pd₄Cu₁-FeNi(OH)₂ as the catalysts. A high urea yield rate of 436.9 mmol/g and FE of 66.4%, as well as ultralong cycling stability of 1000h, are achieved for the Pd₄Cu₁-FeNi(OH)₂ catalysts. They report that Cu is atomically dispersed in Pd and the Cu single-atom alloy promotes the pivotal C-N coupling between *NH₂ and *CO intermediates. The catalytic performance in this work is impressive, while the high performance of Pd₄Cu₁-FeNi(OH)₂ is not well understood. Before further consideration of the publication of this manuscript. Several comments are listed below for the authors' reference.

1. The urea yield rate is significantly improved from 60.4 mmol/g to 436.9 mmol/g by Fe-doping in Ni(OH)₂, which supports the Pd₄Cu₁ nanoparticles for catalyzing C-N coupling. However, the reason for this improvement remains unclear and has not been well studied in this manuscript. Most of the manuscript focuses on the catalyst without doping, which has relatively inferior performance.
2. The authors studied a series of Pd: Cu ratios and found that 4:1 is the best one with atomically dispersed Cu. Actually, in this case, the content of Cu is quite high (~ 20%). How to form Cu single sites at such a high-level Cu concentration. In addition, in their DFT calculations, the model is a doping Cu atom on the Pd surface, in which the Cu content is much less than the one in experiments. What's the actual structure of Pd₄Cu₁ in experiments?
3. In Table 1, Urea can be produced when using HCOOH as the C-source. What is the reaction pathway? How to form *CO for C-N coupling when using HCOOH as the C-source?
4. Lines 349-351, in the DFT calculations, the differential charge density is used to confirm that "*NH₂ prefers to adsorb Cu sites which *CO on Pd sites". How to quantitatively estimate the adsorb strength from differential charge density? Why not use energy to confirm the adsorb site of *CO and *NH₂?
5. In Figure 4d, the *NH₃ formation is energetically more favorable than *NH₂CO (C-N coupling). It seems to contradict the experimental results.

Response to Comments

Reviewer #1 (Remarks to the Author):

The paper "Kinetically matched C–N coupling toward efficient urea electrosynthesis enabled on copper single-atom alloy" represents a high-efficiency urea production rare and high Faradic efficiency. The catalyst Pd₄Cu₁-FeNi(OH)₂ is well characterized and the authors achieved a high catalytic activity and stability in urea electrosynthesis. Despite this appreciable effort, some statements which are not really supported by solid proofs.

Reply: We are grateful to the referee for his/her pertinent evaluation of our work, and appreciate his/her suggestions to help us further improve the quality of our manuscript.

1. Isolated Pd or Cu is avoided during the co-reduction of Pd²⁺ and Cu²⁺ at various molar ratio, but exclusive PdCu single atom alloy. Please clarify the synthetic process.

Reply: We thank the referee for his/her helpful suggestion. The specific synthetic process was listed as follow: Ni(OH)₂ (35.3 mg) nanosheets powder was ultrasonically dispersed in 20 mL DI water for 5 min. Then, K₂PdCl₄ (3.13 mg) and CuCl₂·2H₂O (0.4 mg) were dissolved in the above mixture solution. After that, ice water cooled NaBH₄ solution (1.0 mM, 6 mL) was dropped in the mixture to reduce Pd²⁺ and Cu²⁺ to form Pd₄Cu₁ alloy cluster. After stirring for another 1 h, the final product was collected by centrifugation, washed three times with ethanol and water, and dried in a vacuum oven for 24 h.

To illustrate the avoiding of isolated Cu nanoparticles, we firstly carried out the control experiment to manifest whether solo Cu²⁺ could be reduced with the same amount of NaBH₄ in the presence of Ni(OH)₂ nanosheets. Cu²⁺ could not be reduced to Cu nanoparticles with solo Cu²⁺ ions. While PdCl₄²⁻ ions could be reduced to Pd nanoparticles anchored on Ni(OH)₂ nanosheets (See Supplementary Fig. S5f). The reason could be attributed to lower reduction potential of Cu²⁺/Cu⁰ (0.340 V) than Pd²⁺/Pd⁰ (0.951 V). The result indicates that isolated Cu nanoparticles could not be formed during the co-reduction of PdCl₄²⁻ and Cu²⁺. The formation of PdCu alloy could be ascribed to Cu²⁺ underpotential deposition on Pd as the pre-formed Pd nanocrystals served as the catalytic sites for Cu²⁺ reduction (*Chem. Rev.*, **2016**, *116*, 10414-10472, *Electrochimica Acta*, **2017**, *229*, 415-421). Elemental mapping profiles (Fig. 1c) verify the uniform distribution of Pd and Cu across the particles, suggesting the formation of PdCu alloy structure. Combined with XAFS result (Fig. 1f, Supplementary Table 2), Cu–Cu bond was absent and Pd–Cu bond was resolved in Pd₄Cu₁-Ni(OH)₂ sample. Putting together, we can conclude that Cu is atomically

dispersed in Pd lattice, i.e., Cu single-atom alloy. Notably, we can not completely exclude the possibility of isolated Pd nanoparticles in the final sample.

2. From Fig. 1c, the distribution of Cu and Pd does not match well. It seems this result can not support the formation of PdCu single atom alloy.

Reply: We thank the referee for pointing out this issue. In order to reveal the distribution of Pd and Cu, elemental mapping test with higher resolution was carried out again. As shown in Fig. R1, Pd (green) and Cu (yellow) are uniformly distributed across the nanoparticles, suggesting the formation of PdCu alloy structure. Of particular note, Cu single-atom alloy structure was finally confirmed by EXAFS result (Fig. 1f, Supplementary Table 2), in which Cu–Cu bond was absent and Pd–Cu bond was resolved in Pd₄Cu₁-Ni(OH)₂ sample. Therefore, we conclude that Cu is atomically dispersed in Pd lattice, i.e., Cu single-atom alloy. According to the suggestion, Fig. 1c has been replaced with a new one in the revised manuscript.

Fig. R1 | EDS elemental mapping profile of Pd₄Cu₁-Ni(OH)₂ composite structure.

3. From the R-space and WT counter plot, the spectra of Pd₄Cu is quite consistent with Cu foil. Generally, Pd should induce strain in Cu lattice, resulting the expansion in scattering path of Cu. See *Nano Energy*, 2022, 102, 107704; and *Chem Catalysis* 2021, 1, 1088–1103. Besides, The XAFS spectra of Pd is missing.

Reply: We thank the referee for bringing this to our consideration. We totally agree that alloying Pd in Cu lattice will expand the scattering path of Cu due to larger atomic radius of Pd (202 pm) and Cu (140 pm, See <https://periodictableguide.com/atomic-radius-chart-of-all-elements/>). As shown in Supplementary Table 2, the bond lengths of Cu–Cu and Pd–Pd in Cu foil and Pd foil are 2.54 and 2.74 Å, respectively. Pd–Cu scattering paths derived from Pd K-edge EXAFS and Cu K-edge EXAFS are 2.62 and 2.61 Å in Pd₄Cu₁-Ni(OH)₂, respectively, larger than that of Cu–Cu bond and smaller than that of Pd–Pd bond. The result confirms that alloying Cu in Pd lattice indeed results in an eclectic Pd–Cu bond length in Pd₄Cu₁ clusters.

The provided two papers display the bond lengths of Pd–Cu were 2.45 Å (Pd_{0.7}Cu/CNTs, *Nano Energy*, **2022**, *102*, 107704) and 2.44 ± 0.04 Å (PdCu/Cu₂O, *Chem Catal.*, **2021**, *1*, 1088-1103), respectively. Pd–Cu bond lengths are smaller than that in our work. That is because the bond length of Cu–Cu (about 2.20 Å) in referenced Cu foil is smaller than that in our work (Cu–Cu bond length: 2.54 Å) due to different line stations. Some recent published papers also demonstrate Cu–Cu bond lengths in Cu foil are about 2.5 Å (*Nat. Catal.*, **2022**, *5*, 251-258, *J. Am. Chem. Soc.*, **2017**, *139*, 4486-4492, *Sci. Adv.*, **2017**, *3*, e1701069). The results indicate the conclusion is consistent with that provided papers. The above two paper has been added in Ref. 24, 25 in the revised manuscript.

Pd K-edge XANES spectrum of Pd₄Cu₁-Ni(OH)₂ was also obtained in reference with Pd foil (Fig. R2). The two curves are almost overlapped, indicating metallic Pd feature in Pd₄Cu₁-Ni(OH)₂ sample (Fig. R2a). Pd–Pd (2.70 Å) and Pd–Cu (2.62 Å) bonds are all resolved with CNs of 7.9 and 1.4 in Pd K-edge EXAFS (See Supplementary Table 2, Fig. R2b), respectively. Consistent with Cu case, the fitting curve is almost overlapped with experiment spectrum, validating the reliability of the fitting result (Fig. R2c). Wavelet transforms (WT) analysis of the Pd K-edge EXAFS oscillations of Pd₄Cu₁-Ni(OH)₂ sample resolves Pd–Cu bond (Fig. R2d). According to the suggestion, XAFS of Pd K-edge has been added in Supplementary Fig. 9 in the revised supporting information.

Fig. R2 | (a) Normalized Pd K-edge XANES spectra of Pd₄Cu₁ in reference with Pd foil, (b) k^3 -weighted Fourier-transform Pd K-edge EXAFS spectra, (c) the experimental Pd K-edge EXAFS spectrum (red circle) and the fitting curve (black line) of Pd₄Cu₁. (d) Wavelet transforms of the k^2 -weighted Pd K-edge EXAFS signals for the high-coordination shells in reference with Pd foil.

4. *All the characterization, including in-situ Raman, only indicates the Pd₄Cu₁-Ni(OH)₂ is a superior urea-electrosynthesis catalyst, which is far from the topic of kinetic match C-N coupling.*

Reply: We thank the referee for bringing this to our attention. In order to confirm the matched kinetics of CO₂RR and NO₃RR boosts urea electrosynthesis, the kinetics of CO₂RR and NO₃RR was systematically regulated. It is well known that the kinetics of CO₂RR and NO₃RR are closely associated with CO₂ partial pressure or the concentrations of NO₃⁻. The lower partial pressure of CO₂ in CO₂RR or concentrations of NO₃⁻ in NO₃RR reduces the reduction kinetics. As shown in Fig. R3a, urea yield rates and urea FEs of Pd₄Cu₁-Ni(OH)₂ decrease with CO₂ partial pressure, suggesting that the reduced CO₂RR kinetics indeed lowers urea production. Fig. R3b shows the relationship of urea yield rates and urea FEs with the concentrations of NO₃⁻. Similarly, urea yield rates and urea FEs decrease with the concentrations of NO₃⁻ in the range of 0.1 M-0.02 M, and the optimal urea yield rate is obtained with 0.1 M of NO₃⁻. Fig. R3a and R3b confirm that urea yield rates and urea FEs are indeed related to the kinetics of CO₂RR and NO₃RR.

To further verify the matched kinetics of CO₂RR and NO₃RR boosts urea formation, the yield rates of CO and NH₃ in solo CO₂RR and NO₃RR were obtained. Notably, the real formation kinetics of *CO and *NH₂ (the key C- and N-intermediates for C-N coupling) are hardly to obtain in the co-reduction, and replaced with the yield rates of CO and NH₃. As shown in Fig. R3c, NH₃ yield rates are increased for Pd-Ni(OH)₂, Pd₁Cu₁-Ni(OH)₂ and Pd₄Cu₁-Ni(OH)₂. However, CO yield rates show an opposite trend for the three samples. More importantly, the ratios of NH₃:CO increased from 0.12, 0.66 to 1.48, and the value of 1.48 is approaching the theoretical value (*NH₂:*CO= 2 in urea) for Pd₄Cu₁-Ni(OH)₂. Accordingly, urea yield rates and urea FE all increase from 2.3 mmol g⁻¹ h⁻¹, 6.6%, 2.6 mmol g⁻¹ h⁻¹, 6.9% to 18.8 mmol g⁻¹ h⁻¹, 76.2%. All the above results confirm the matched kinetics of CO₂RR and NO₃RR really boosts urea electrosynthesis. Further increasing NH₃ and CO yield rates (NH₃:CO= 3.04, Fig. R3c) for Pd₄Cu₁-FeNi(OH)₂ sample results in increased urea yield rate (34.1 mmol g⁻¹ h⁻¹) but decreased urea FE (59.3%), verifying this conclusion. Single-atom Cu in Pd lattice tunes the kinetics of CO₂RR and NO₃RR to approach the theoretical value in urea formation. The matched kinetics of CO₂RR and NO₃RR is responsible for the ultrahigh urea yield rate and urea FE in Pd₄Cu₁-Ni(OH)₂. According to the suggestion, the newly obtained results have been added in Fig. 3f-3h in the revised manuscript.

Fig. R3 | Urea yield rates and urea FEs of Pd₄Cu₁-Ni(OH)₂ assessed with (a) varied CO₂ partial pressure in the mixture of CO₂ + Ar or (b) concentrations of NO₃⁻ in the electrolyte. (c) Production rates of CO and NH₃ in solo CO₂RR and NO₃RR at -0.5 V, and the corresponding ratios of NH₃:CO.

5. The authors state that the introduction of Fe to forming Pd₄Cu-FeNi(OH)₂ could further improving the catalytic performance by 7-fold in urea production. The authors should affirm that whether the Pd₄Cu is the active site for urea electro-synthesis. The authors did not uncover the catalytic mechanism of the urea production over Pd₄Cu-Ni(OH)₂ or Pd₄Cu-FeNi(OH)₂.

Reply: We thank the referee for bringing this to our consideration. According to the suggestion, some control experiments were carried out to affirm Pd₄Cu₁ clusters were the active sites for urea electro-synthesis. As we could not obtain mono-dispersed Pd₄Cu₁ clusters with comparable size, we evaluated urea electro-synthesis performance of Ni(OH)₂ and FeNi(OH)₂ nanosheets. As shown in Fig. R4a, urea yield rates and urea FEs at -0.5 V are 0.7 mmol g⁻¹ h⁻¹, 5.5% and 2.7 mmol g⁻¹ h⁻¹, 23.4% in H-type cell for Ni(OH)₂ and FeNi(OH)₂ nanosheets, respectively. Obviously, both urea yield rates and urea FEs for Ni(OH)₂ and FeNi(OH)₂ nanosheets are much lower than that of Pd₄Cu₁-Ni(OH)₂ sample (18.8 mmol g⁻¹ h⁻¹, 76.2%) at -0.5 V. The results indicate that the carriers of Ni(OH)₂ and FeNi(OH)₂ in the composite catalysts are not the active center for urea formation, and the Pd₄Cu₁ clusters are the real active sites for C-N coupling toward urea electro-synthesis (*EES Catal.*, **2023**, *1*, 45-53).

To uncover the role of Ni(OH)₂ carriers on urea electro-synthesis, solo CO₂RR and NO₃RR were carried out using Ni(OH)₂ or FeNi(OH)₂ nanosheets as catalysts. As shown in Fig. R4b, no reduction product, e.g., CO in CO₂RR is detected both for Ni(OH)₂ and FeNi(OH)₂ nanosheets, suggesting that Ni(OH)₂ and FeNi(OH)₂ nanosheets are inert for CO₂RR. Then, solo NO₃RR performance was also evaluated, NH₃ yield rate and FE at -0.5 V are 5.6 mmol g⁻¹ h⁻¹ and 10.9% for Ni(OH)₂ nanosheets. Notably, NH₃ yield rate and NH₃ FE are much lower than that of Pd₄Cu₁-Ni(OH)₂ (171.0 mmol g⁻¹ h⁻¹, 64.9%), suggesting that Ni(OH)₂ carrier has a weak catalytic capacity for NO₃RR. The result also confirms that Pd₄Cu₁ clusters are the real active sites for CO₂RR, NO₃RR and C-N coupling.

Then, solo NO₃RR performance was also assessed for FeNi(OH)₂ nanosheets to decode the role of Fe doping in Ni(OH)₂ on the greatly enhanced urea electro-synthesis

for Pd₄Cu₁-FeNi(OH)₂. As shown in Fig. R4c, NH₃ yield rate increases from 5.6 to 10.9 mmol g⁻¹ h⁻¹ at -0.5 V, indicating that Fe doping in Ni(OH)₂ facilitates the deep reduction of NO₃RR to NH₃ (*Nanoscale*, **2023**, *15*, 204-214). As such, the produced *NH₂ on FeNi(OH)₂ nanosheets in Pd₄Cu₁-FeNi(OH)₂ interface could partly facilitate C–N coupling toward urea electrosynthesis.

Fig. R4 | The comparison of FeNi(OH)₂ and Ni(OH)₂ nanosheets in (a) urea electrosynthesis, (b) solo NO₃RR and (c) CO₂RR. (d) Production rates of CO and NH₃ in solo CO₂RR and NO₃RR at -0.5 V, and the corresponding ratios of NH₃:CO.

Apart from the slightly enhanced NO₃RR on FeNi(OH)₂ nanosheets, the dissociation of H-OH to produce active H atoms is also enhanced on Pd₄Cu₁-FeNi(OH)₂, which is vital important for the deoxyreduction process (NO₃⁻ → *NH₂, CO₂ → *CO). Previous reports have demonstrated that M-Ni(OH)₂ interface (M= Pt, Ru) facilitated water dissociation by forming M···H-O²⁻···Ni²⁺ interaction (*Science*, **2011**, *334*, 1256-1260, *Nat. Mater.*, **2012**, *11*, 550-557). And high-valence state Fe³⁺ doping in Ni(OH)₂ further accelerated this process by forming stronger Fe³⁺···O²⁻-H interaction (*J. Alloys Comp.*, **2020**, *823*, 153790). The current density of Pd₄Cu₁-FeNi(OH)₂ in LSV curves (Fig. R5a) is greatly enhanced than that of Pd₄Cu₁-Ni(OH)₂. Tafel plots (Fig. R5b) also indicate an enhanced hydrogen evolution kinetics for Pd₄Cu₁-FeNi(OH)₂. The results suggest that the dissociation of H-OH to produce active H atoms is indeed enhanced, and more active H atoms facilitate urea formation.

To further confirm which factor dominates the greatly enhanced urea yield rate for Pd₄Cu₁-FeNi(OH)₂ than that of Pd₄Cu₁-Ni(OH)₂, we further carried out electrochemical C–N coupling using D₂O as H-source. The dissociation rate of D-OD

and D transfer process is slower than that of H-OH due to isotope effect, which would result in declined urea yield rate (*Angew. Chem. Int. Ed.*, **2020**, *59*, 21170-21175). As shown in Fig. R6. Urea yield rate and urea FE are synchronously declined from 34.1 mmol g⁻¹ h⁻¹, 59.3% to 5.7 mmol g⁻¹ h⁻¹ and 38.0% when using D₂O as H-source. Urea yield rate is about one sixth with D₂O as H-source than that with H₂O. Therefore, we can conclude that the enhanced water dissociation to produce active H atoms is response for the enhanced urea yield rate for Pd₄Cu₁-FeNi(OH)₂.

In conclusion, the reason for the greatly enhanced urea yield rate after Fe doping in Ni(OH)₂ carrier for Pd₄Cu₁-FeNi(OH)₂ composite catalyst can be summarized as follows: 1. Fe³⁺ doping in Ni(OH)₂ carrier enhances water dissociation to produce more active H atoms on Pd₄Cu₁ clusters, then the yield rates of *NH₂ and *CO are enhanced to boost urea formation (Fig. R4d). 2. More *NH₂ on FeNi(OH)₂ nanosheets itself facilitates C-N coupling in Pd₄Cu₁-FeNi(OH)₂ interface. According to the suggestion, the role of Fe-doped Ni(OH)₂ on the greatly enhanced urea formation has been highlighted in the revised manuscript and the newly obtained data has been added in Fig. 5b, 5c in the revised manuscript and Supplementary Fig. 44, 45 in the revised supporting information.

Fig. R5 | (a) LSV curves and (b) the corresponding Tafel plots of Pd₄Cu₁-Ni(OH)₂ and Pd₄Cu₁-FeNi(OH)₂ composite samples.

Fig. R6 | Urea yield rate and FE for Pd₄Cu₁-FeNi(OH)₂ at -0.5 V using H₂O or D₂O as H-source.

6. Some expression are wrong, e.g. Na(OH)₂

Reply: We thank the referee for his/her careful suggestion. According to the suggestion, we have re-checked our manuscript and corrected the mistakes in the revised manuscript.

For these reasons, I cannot recommend the paper in this form for publication in Nat. Commun.

Reply: We thank again the referee for his/her valuable suggestions to improve the quality of our work. We hope our revision have solved the concerns of the referee and reached the quality for publication on *Nature Communications*.

Reviewer #2 (Remarks to the Author):

The authors report a single-atom copper-alloyed Pd catalyst (Pd₄Cu₁) that can achieve highly efficient C–N coupling toward urea electrosynthesis. The synthesized Pd₄Cu₁-FeNi(OH)₂ composite catalyst achieves a recorded urea yield rate of 436.9 mmol g_{cat}⁻¹ h⁻¹ and FE of 66.4%, as well as ultralong cycling stability of 1000 h. There are several issues that need to be addressed and hence I recommend a major revision before being considered for publication.

Reply: We are grateful to the referee for his/her highly positive evaluation of our work, and appreciate his/her suggestions to help us further improve the quality of our manuscript.

1. What was the rationale behind going to single atom copper catalyst?

Reply: We thank the referee for his/her insightful question. The reason to choose Cu single-atom catalyst for urea electrosynthesis can be summarized as follows: 1. Single-atom Cu in Pd host can form Pd–Cu sites and avoid the formation of Cu–Cu bond. The adsorption of CO₂ on Pd–Cu sites has a totally different configuration with that on Cu–Cu sites. The former possesses larger adsorption energy, manifesting that Cu single-atom alloy can strength CO₂ adsorption on the catalyst surface (*J. Am. Chem. Soc.*, **2017**, *139*, 4486–4492). 2. Our previous work has demonstrated that Cu single-atom catalyst facilitates the deep reduction of NO₃RR to NH₃, while N–N coupling to yield N₂ in NO₃RR is preferred on Cu–Cu sites, which declines the urea yield rate and urea FE (*ChemSusChem*, **2022**, *15*, e2022002). 3. Cu single-atom in Pd host can achieve maximum Pd–Cu charge polarization, which facilitates to stabilize the key C- (*CO) and N-intermediates (*NH₂) to promote C–N coupling toward urea formation. 4. NO₃RR usually possesses a larger kinetics as the solubility of NO₃⁻ is greatly larger than that of CO₂ in electrolyte. Hence, by decreasing the Cu doping amount to form Cu single-atom alloy, the kinetics of CO₂RR and NO₃RR are matched on Pd₄Cu₁ clusters, which boosts urea yield rate and urea FE. For the above reasons, Cu single-atom alloy was chosen for urea electrosynthesis and the results in the manuscript have confirmed that urea yield rate and urea FE was indeed greatly enhanced on Pd₄Cu₁-Ni(OH)₂ composite catalyst.

2. Were the experimental results compared with planar Cu, planar Ni and planar Pd? If not, please perform the needed experiments and show the trends of urea selectivity, FE and current density.

Reply: We thank the referee for his/her meaningful suggestion. According to the suggestion, Pd nanosheets (*Nature*, **2019**, *574*, 81-85), Cu nanosheets (*EcoMat.*, **2023**, *5*, e12334) and Ni nanosheets (*Angew. Chem. Int. Ed.*, **2016**, *55*, 693-697) were synthesized. XRD patterns (Fig. R7a) and TEM images (Fig. R7b-7d) confirm the successful synthesis of the three nanosheets. Urea yield rates and urea FEs at –0.5 V

are 4.8, 1.0, 0 mmol g⁻¹ h⁻¹, 19.3%, 3.6% and 0% for Ni nanosheets, Pd nanosheets and Cu nanosheets, respectively. The urea partial current densities for Ni nanosheets, Pd nanosheets and Cu nanosheets are 0.18, 0.03 and 0 mA cm⁻² at -0.5 V (Fig. R7f). According to the suggestion, the newly obtained data has been added in Supplementary Fig. 38 in the revised supporting information.

Fig. R7 | (a) XRD patterns of Cu nanosheets and Ni nanosheets. TEM images of (b) Cu nanosheets, (c) Ni nanosheets and (d) Pd nanosheets. (e) Urea yield rates and urea FEs, (f) urea partial current density.

3. In line 71, please mention the total loading on the GDE.

Reply: We thank the referee for his/her helpful suggestion. The catalyst loading amount for Pd₄Cu₁-Ni(OH)₂ and Pd₄Cu₁-FeNi(OH)₂ on the GDE are 0.1 and 0.025 mg cm⁻², respectively. According to the suggestion, the catalyst loading amount has been added in the revised manuscript.

4. Fig 1a shows the TEM image which is too unclear to analyze and conclude the uniform distribution. Moreover, they seem to be more like clusters instead of single atoms dispersed on the lattice. Please refer to this article as a reference to show TEM images for single atoms dispersion. Link: <https://www.frontiersin.org/articles/10.3389/fchem.2021.717201/full>.

Reply: We thank the referee for his/her meaningful suggestion. According to the suggestion, aberration-corrected high-angle annular dark-field scanning TEM (HAADF-STEM) was used to re-characterize Pd₄Cu₁-Ni(OH)₂ sample. As shown in Fig. R8, uniform Pd₄Cu₁ clusters with average diameter of 3.5 ± 0.1 nm are anchored on Ni(OH)₂ nanosheets. In order not to misunderstand, the schematic diagram of Pd₄Cu₁-Ni(OH)₂ was inserted in Fig. R8. Notably, Cu single-atom alloy feature was confirmed by XAFS, not by HAADF-STEM. According to the suggestion, Fig. 1a has been replaced with a new one and the above literature has been added in Ref. 19 in the revised manuscript.

Fig. R8 | Aberration-corrected HAADF-STEM image of Pd₄Cu₁-Ni(OH)₂ composite structure.

5. Fig 1b is not clear at all and authors have pointed to a measurement and referred to it as diameter, whereas nothing spherical is seen in this image. Please specify how you measured that distance *d* and what it denotes. Moreover, the image quality is very poor to analyze and conclude anything.

Reply: We thank the referee for his/her helpful question. In order to acquire clear outline and lattice fringe of Pd₄Cu₁ clusters, aberration-corrected high-angle annular dark-field scanning TEM (HAADF-STEM) was used to re-characterize the Pd₄Cu₁-Ni(OH)₂ sample. As shown in Fig. R9, spherical like Pd₄Cu₁ clusters are observed and the lattice distance of 0.22 nm can be attributed to (111) plane of Pd₄Cu₁ cluster. The size of the Pd₄Cu₁ clusters was obtained by counting the diameters of more than 100 particles. According to the suggestion, Fig. 1b has been replaced by the newly obtained data in the revised manuscript.

Fig. R9 | Aberration-corrected HAADF-STEM image of Pd₄Cu₁-Ni(OH)₂ composite structure.

6. What does Fig 1c explains? Please explain the first and fourth part of figure 1c properly.

Reply: We thank the referee for his/her insightful question. In order to acquire more clear Pd and Cu distribution, elemental mapping profiles with higher resolution was obtained. As shown in Fig. R10, Pd and Cu are uniformly distributed across the clusters, which is confirmed by the merged images (*Adv. Mater.*, **2017**, 29, 1700769), suggesting the PdCu alloy structure. According to the suggestion, Fig. 1c has been replaced with the newly obtained data in the revised manuscript.

Fig. R10 | EDS elemental mapping profile of Pd₄Cu₁-Ni(OH)₂ composite structure.

7. In line 112, authors mention that XPS and ICP-MS show the same results. Why were they both performed to validate the same results? What other insights did you get from each of them besides this?

Reply: We thank the referee for his/her meaningful question. ICP-MS measurement

can determine the overall constituent of the sample by thoroughly dissolving of the sample. However, XPS can give a composition analysis of the surface of the sample as X-ray has a certain depth of detection (lower than 2 nm). Pd₄Cu₁ clusters possess an average size of 3.5 ± 0.1 nm, ICP-MS combined with XPS result confirm that the constituent of Pd:Cu are comparable between the surface and bulk phase of Pd₄Cu₁ clusters. Beyond that, the two characterization results confirm each other, which also indicates that the characterization of the components is credible. Besides, XPS test also give the refined Cu 2p, Ni 2p and Pd 3d spectra (Fig. 1d and Supplementary Fig. 7). The binding energy of Cu 2p 3/2 and Cu 2p 1/2 for Pd₄Cu₁ clusters is shifted to higher region than Cu clusters (Fig. 1d), confirming the electron transfer from Cu to adjacent Pd atoms.

8. What insights did wave transform analysis give for this study? The contours in Fig 1h don't explain or signify anything. Please put a clearer figure and explain what it signifies.

Reply: We thank the referee for his/her insightful question. In wavelet transforms (WT) spectra, the abscissa represents the wave vector number (k), which is directly related to the atomic number. The abscissa can be used to determine the type of chemical bonds. The ordinate represents the bond length of the chemical bonds. Therefore, we can directly obtain the bond lengths and type information from the coordinates of the strongest signal in the wavelet transform image (Fig. R11). The coordinate values are (7.35, 2.22), (5.20, 1.50) and (7.50, 2.19) for Pd₄Cu₁, CuO and Cu foil, respectively, revealing Pd–Cu bond (bond length: 2.22 Å), Cu–O bond (bond length: 1.50 Å) and Cu–Cu bond (bond length: 2.19 Å) for Pd₄Cu₁, CuO and Cu foil. According to the suggestion, Fig. 1h has been replaced with a clearer one and bond information was highlighted in the revised manuscript.

Fig. R11 | Wavelet transforms of the k^2 -weighted Cu K-edge EXAFS signals for the high-coordination shells in reference with Cu foil and CuO.

9. Perform the EXAFS for Pd foil and Ni foil as well to give a clear comparison (Fig 1f).

Reply: We thank the referee for his/her meaningful suggestion. According to the suggestion, EXAFS spectra of Pd foil and Ni foil were obtained. As shown in Fig. R12, Pd–Pd bond and Ni–Ni bond are resolved with bond length located at 2.45 Å and 2.18 Å, respectively. According to the suggestion, the newly obtained data has been added in Fig. 1f in the revised manuscript.

Fig. R12 | k^3 -weighted Fourier-transform Cu K-edge, Pd K-edge and Ni K-edge EXAFS spectra.

10. In fig 2b, show urea current density as well.

Reply: We thank the referee for his/her useful suggestion. As shown in Fig. R13, the urea partial current densities (j_{urea}) at -0.5 V for $\text{Pd}_x\text{Cu}_1\text{-Ni(OH)}_2$ are 0.04, 0.13, 0.19, 0.46, 0.68, 0.49, 0.38 and 0.19 mA cm^{-2} for Ni(OH)_2 , $\text{Pd}_1\text{Cu}_1\text{-Ni(OH)}_2$, $\text{Pd}_2\text{Cu}_1\text{-Ni(OH)}_2$, $\text{Pd}_3\text{Cu}_1\text{-Ni(OH)}_2$, $\text{Pd}_4\text{Cu}_1\text{-Ni(OH)}_2$, $\text{Pd}_5\text{Cu}_1\text{-Ni(OH)}_2$, $\text{Pd}_6\text{Cu}_1\text{-Ni(OH)}_2$ and Pd-Ni(OH)_2 , respectively. According to the suggestion, urea partial current densities have been added in Supplementary Fig. 14a in the revised supporting information.

Fig. R13 | Urea partial current densities for $\text{Pd}_x\text{Cu}_1\text{-Ni(OH)}_2$ composite samples and Ni(OH)_2 at -0.5 V in H-type cell.

11. In Fig 2c and 2d provide the urea current density data as well.

Fig. R14 | Urea partial current densities for $\text{Pd}_4\text{Cu}_1\text{-Ni(OH)}_2$ composite samples (a) in H-type cell and (b) in GDE.

Reply: We thank the referee for his/her useful suggestion. As shown in Fig. R14a, the urea partial current densities (j_{urea}) for $\text{Pd}_4\text{Cu}_1\text{-Ni(OH)}_2$ in H-type cell are 0.08, 0.05, 0.07, 0.18, 0.68 and 0.53 mA cm^{-2} at -0.1 , -0.2 , -0.3 , -0.4 , -0.5 and -0.6 V,

respectively (Fig. R14a). The urea partial current densities for Pd₄Cu₁-Ni(OH)₂ in GDE are 0.23, 0.32, 0.37, 0.52, 2.26 and 1.9 mA cm⁻² at -0.1, -0.2, -0.3, -0.4, -0.5 and -0.6 V, respectively (Fig. R14b). Urea partial current densities in GDE are greatly increased than that in H-type cell. According to the suggestion, the current densities have been added in Supplementary Fig. 14b and 14c in the revised supporting information.

12. In Fig 2e, why hasn't the urea yield rate been shown till 100 h. This is a multiple-product reaction unlike HER, hence stable total current density for 100h doesn't necessarily mean stable urea current for 100 hours. Also show urea current density along with yield rate for 100h to comment on the stability.

Reply: We thank the referee for bringing this to our attention. We totally agree that the stable current density in urea electrosynthesis do not mean stable urea yield rate in long-term stability test. According to the suggestion, we also acquired urea yield rates during the 100 h stability test. As shown in Fig. R15, urea yield rates slowly decline from 18.7 mmol g⁻¹ h⁻¹ at the initial 2 h to 12.8 mmol g⁻¹ h⁻¹ at 100 h. Urea partial current density (j_{urea}) changes from 0.48 mA cm⁻² at the initial 2 h to 0.38 mA cm⁻² at 100 h. The result indicates that the urea yields rates are stable. According to the suggestion, the newly obtained data has been added in Fig. 2e in the revised manuscript.

Fig. R15 | Cycling stability of Pd₄Cu₁-Ni(OH)₂ in urea electrosynthesis assessed in H-type cell.

13. In figure 2f, show the urea current density and yield rate for 360 hours to comment on the stability of the GDE.

Reply: We thank the referee for his/her insightful suggestion. According to the suggestion, the urea partial current densities (j_{urea}) are 2.8 mA cm⁻² in the initial 20 h and then changed to 2.6 mA cm⁻² at 380 h in GDE (Fig. R16). Accordingly, urea yield rate changes from 72.3 mmol g⁻¹ h⁻¹ at 20 h to 66.2 mmol g⁻¹ h⁻¹ at 380 h, suggesting the stability of Pd₄Cu₁-Ni(OH)₂ composite catalyst in GDE. According to the suggestion, the yield rates and urea partial current densities have been added in Fig. 2f in the revised manuscript.

Fig. R16 | Cycling stability of Pd₄Cu₁-Ni(OH)₂ catalyst in urea electrosynthesis assessed in GDE.

14. In figures 3a, b, c, please mention the current density of each of the products.

Reply: We thank the referee for his/her useful suggestion. According to the suggestion, partial current densities of the possible products are shown in Fig. R17. The newly obtained data has been added in Supplementary Fig. 28 in the revised supporting information.

Fig. R17 | The partial current densities of the possible products for (a) Pd-Ni(OH)₂, (b) Pd₁Cu₁-Ni(OH)₂ and (c) Pd₄Cu₁-Ni(OH)₂ composite catalysts.

15. Perform the theoretical calculations for pure Cu and Ni as well to get a thorough comparison.

Reply: We thank the referee for his/her insightful suggestion. According to the suggestion, the energy profiles of each elementary step in urea production for Cu(111) and Ni(111) planes were also obtained. As shown in Fig. R18 and Table R1, the energy barriers of $*NO \rightarrow *HNO$ process for Cu(111) and Ni(111) planes are 0.44 and 1.09 eV, respectively, lower than that of Pd(111) (1.15 eV). The results indicate that Cu(111) plane is more active than Pd(111) and Ni(111) planes in NO_3RR , and Cu single-atom in Pd host really facilitates NO_3RR performance. Then, the energy barrier of $*NH_2 \rightarrow *NH_2CO$ were also obtained, which are 0.62 and 0.31 eV for Cu(111) and Ni(111) planes, higher than that Pd(111) (0.19 eV) and $Cu_1Pd(111)$ (0.07 eV). The theoretical calculation results well explain the tendency of urea formation for Pd nanosheets, Cu nanosheets and Ni nanosheets (See response 2 to reviewer 2). Cu(111) planes possess the lower energy barrier of $*NO \rightarrow *HNO$ process and the larger barrier of $*NH_2 \rightarrow *NH_2CO$, which makes Cu(111) planes are preferable for NO_3RR to NH_3 , not for urea formation. While Ni(111) planes possess eclectic energy barrier of $*NO \rightarrow *HNO$ and $*NH_2 \rightarrow *NH_2CO$ process. As such, Ni nanosheets deliver better urea yield rate. According to the suggestion, the newly obtained data for Cu(111) and Ni(111) planes has been added in Fig. 4d in the revised manuscript and Supplementary Table 6 in the revised supporting information.

Fig. R18 | Energy profiles of each elementary step in NO_3RR with C–N coupling toward urea synthesis catalyzed by $Cu_1Pd(111)$, Pd(111), Cu(111) and Ni(111) planes.

Table R1. The Gibbs free energy change (ΔG , eV) of reaction for urea formation on $Cu_1Pd(111)$, Pd(111), Cu(111) and Ni(111) at 0 V (vs. RHE).

Intermediates	$Cu_1Pd(111)$	Pd(111)	Cu(111)	Ni(111)
$*NO_2$	0	0	0	0
$*NO_2H$	0.39	0.39	0.57	0.50
$*NO$	-1.87	-2.17	-0.91	-1.10
$*HNO$	-1.13	-1.02	-0.46	-0.01
$*H_2NO$	-1.37	-1.60	-1.46	-1.56
$*H_2NOH$	-1.56	-1.96	-1.34	-1.68
$*NH_2$	-3.44	-3.55	-3.55	-3.04
$*NH_2CO$	-3.37	-3.36	-2.93	-2.72
$*NH_2CONH_2$	-9.81	-9.82	-9.57	-9.50

16. Was XAFS just done to confirm the fine structure of Pd? It looks like it was already established using other characterizations performed.

Reply: We thank the referee for his/her meaningful question. Apart from the fine structure of Pd, synchrotron radiation-based X-ray absorption fine structure (XAFS) spectroscopy results also provide extra three important findings. 1. As shown the insert in Fig. R19e, the pre-edge of Cu K-edge XANES spectra between 8975 and 8995 eV for Pd₄Cu₁ clusters is approaching but slightly lower than that of Cu foil, suggesting the valence of Cu^{δ+} in Pd₄Cu₁ is 0<δ<2, confirming charge polarization between Cu and Pd atoms (Cu^{δ+} → Pd^{δ-}). 2. From Cu K-edge EXAFS spectra (Fig. R19f) and fitting results (Table R2), Cu–Cu bond is absent and Pd–Cu bond is resolved in Pd₄Cu₁ clusters, suggesting atomically dispersed Cu atoms in Pd host, i.e., Cu single-atom alloy. It should be noted that Cu single-atom alloy structure can only be confirmed by XAFS characterizations. 3. The detailed bond type, bond length (R) and coordination number (CN) can be derived from EXAFS spectra, Cu–O (R= 2.05 Å, CN= 3.05) and Cu–Pd bonds (R= 2.61 Å, CN= 10.71) in Cu EXAFS (Fig. 1f), Pd–Cu (R= 2.62 Å, CN= 1.35) and Pd–Pd (R= 2.70 Å, CN= 7.87) in Pd EXAFS (Supplementary Fig. 9).

Fig. R19 | (a) HAADF-STEM image, (b) HRTEM image, (c) EDS elemental mapping profile of Pd₄Cu₁-Ni(OH)₂ composite structure. (d) Cu 2p spectra of Pd₄Cu₁-Ni(OH)₂, Pd₁Cu₁-Ni(OH)₂ and Pd-Ni(OH)₂. (e) Normalized Cu K-edge XANES spectra of

Pd₄Cu₁ clusters in reference with Cu foil and CuO, (f) k^3 -weighted Fourier-transform Cu K-edge EXAFS spectra, (g) the experimental Cu K-edge EXAFS spectrum (red circle) and the fitting curve (black line) of Pd₄Cu₁. (h) Wavelet transforms of the k^2 -weighted Cu K-edge EXAFS signals for the high-coordination shells in reference with Cu foil and CuO. The inset in a shows schematic diagram of Pd₄Cu₁-Ni(OH)₂.

Table R2. EXAFS fitting parameters at the Pd Cu K-edge for various samples ($S_0^2=1.0$).

Sample	Path	N^a	$R(\text{\AA})^b$	$\sigma^2(\text{\AA}^2)^c$	$\Delta E_0(\text{eV})^d$	R factor
Cu foil	Cu-Cu	12.00	2.54	0.0098	4.21	0.0076
CuO	Cu-O	6.00	1.95	0.0061	7.45	0.0163
	Cu-Cu	7.92	2.89	0.0153	2.22	
Pd foil	Pd-Pd	11.66	2.74	0.0051	-6.20	0.0033
Sample Cu	Cu-O	3.05	2.05	0.0157	9.07	0.0191
	Cu-Pd	10.71	2.61	0.0120	-5.09	
Sample Pd	Pd-Cu	1.35	2.62	0.0040	-8.64	0.0067
	Pd-Pd	7.87	2.70	0.0102	-6.15	

17. In line 167, authors conclude that alloying Cu single atoms in Pd lattice boosts urea electrosynthesis performance. Was dispersion of Pd atoms in Cu lattice tried to thoroughly conclude that the former arrangement performs better?

Reply: We thank the referee for his/her insightful suggestion. According to the suggestion, Pd atoms in Cu lattice, e.g., Pd₁Cu₆-Ni(OH)₂, Pd₁Cu₄-Ni(OH)₂, Pd₁Cu₂-Ni(OH)₂ composite catalysts were synthesized. TEM images (Fig. R20a-R20c) show that Pd₁Cu₆, Pd₁Cu₄ and Pd₁Cu₂ clusters are uniformly anchored on Ni(OH)₂ nanosheets. Urea electrosynthesis performance was also assessed at -0.5 V. As shown in Fig. R20d, urea yield rates and urea FEs are 0.77, 2.03, 1.39 mmol g⁻¹ h⁻¹ and 8.7%, 17.3%, 9.6% for Pd₁Cu₆-Ni(OH)₂, Pd₁Cu₄-Ni(OH)₂ and Pd₁Cu₂-Ni(OH)₂, respectively. Urea yield rates and urea FE are all lower than that of Pd₄Cu₁-Ni(OH)₂ composite sample, suggesting that Cu single-atom in Pd host boosts urea electrosynthesis. According to the suggestion, the newly obtained data has been added in Supplementary Fig. 12 in the revised supporting information.

Fig. R20 | (a-c) TEM images of Pd₁Cu₂-Ni(OH)₂, Pd₁Cu₄-Ni(OH)₂ and Pd₁Cu₆-Ni(OH)₂ composite samples. (d) Urea yield rates and FEs of the three samples at -0.5 V.

18. In line 171, authors mention how the urea FE and yield rates vary with potential. Please explain the reason for observed values as they don't seem to fall in a particular trend.

Reply: We thank the referee for bringing this to our attention. As shown in Fig. 2c, urea yield rates are 3.4, 1.5, 3.2, 3.8, 18.8 and 9.2 mmol g_{cat.}⁻¹ h⁻¹ at -0.1, -0.2, -0.3, -0.4, -0.5 and -0.6 V. Urea FEs are 14.0%, 14.0%, 16.0%, 31.1%, 76.2% and 33.8%. Urea yield rates and urea FEs synchronously increase as the applied potential shifted from -0.2 V to -0.5 V. The reason can be attributed to the enhanced CO₂RR and NO₃RR ability at more negative potential, which produces more *NH₂ and *CO intermediates for C-N coupling, and the formation kinetics of *NH₂ and *CO are matched. When the applied potential further shifted to -0.6 V, NH₃ yield rates are accompanied by the greatly accelerated hydrogen evolution. However, CO yield rates increase slowly restricted by mass transfer kinetics of CO₂ in electrolyte (*Appl. Catal. B: Environ.*, **2018**, 232, 391-396), which leads to unmatched kinetics of CO₂RR and NO₃RR (see Fig. R3c and Supplementary Fig. 29). Therefore, urea yield rate and FE are synchronously declined, the volcano-shape of urea yield rates are consistent with the previous reported results (*Nat. Sustain.*, **2021**, 4, 868-876, *Angew. Chem. Int. Ed.*, **2023**, 62, e2022109). Anomalously, urea yield rate at -0.1 V is higher than that at -0.2 V. CO and NH₃ yield rates at -0.1 V are low enough (Supplementary Fig. 29b). We

speculate that the possible reason may be due to the different reaction pathway, and the real mechanism is still on the way.

19. *Metals like Fe, Co, Ni, Cu, Zn were chosen to form single metal alloys. Please explain the rationale behind choosing these metals.*

Reply: We thank the referee for his/her meaningful question. The key C- and N-intermediates are *CO and *NH₂ in electrochemical coupling of CO₂ and NO₃⁻ toward urea electrosynthesis. Obviously, urea yield rate and urea FE are determined by the solo CO₂RR and NO₃RR activity and/or the adsorption strength of *CO and *NH₂. For CO₂RR, previous work has revealed Pd catalysts tend to form CO, but CO binds too strong on Pd catalysts, which is not good for the subsequent C–N coupling process. Alloying transition metals (e.g., Fe, Co, Ni, Cu, Zn) can help to reduce adsorption strength of *CO on Pd catalysts surface (*Nat. Catal.*, **2022**, 5, 251-258). For NO₃RR, transition metals like Fe, Co, Ni, Cu are active elements for electrochemical NO₃RR toward NH₃ synthesis (*Nat. Energy*, **2020**, 5, 605-613). For the above two reasons, metals like Fe, Co, Ni, Cu, Zn are chosen to form single-metal alloys for urea electrosynthesis. The results indicate that Cu single-atom alloy displays the best urea electrosynthesis performance.

20. *FE of 69% at -0.4V vs RHE has already been achieved with PdCu bimetallic catalysts. Please refer <https://pubs.rsc.org/en/content/articlelanding/2023/ey/d2ey00038e> and explain what new insights does your work offer?*

Reply: We thank the referee for his/her insightful question. The mentioned paper (*EES Catal.*, **2023**, 1, 45-53) reported PdCu alloy loaded on bacterial cellulose derived carbon framework, which delivered urea yield rate of 12.7 mmol g⁻¹ h⁻¹ at –0.5 V and optimal urea FE of 69.1±3.8% at –0.4 V. The theoretical calculations revealed that the alloying catalyst provided Pd and Cu dual active sites with favored internal electron transferability, enabling generation of key *NO₂ and *CO₂ intermediates to facilitate C–N coupling reaction for urea synthesis. In spite of the similar PdCu alloy promoting urea electrosynthesis, our work provides a totally different viewpoint in urea electrosynthesis. The novelty is listed as follow: 1. The performance of urea electrosynthesis is determined by the kinetics of CO₂RR and NO₃RR. The matched kinetics of CO₂RR and NO₃RR can be achieved by alloying Cu single-atom in Pd lattice and Cu doping amount. 2. In our work, *in-situ* Raman spectroscopy test reveals a totally different reaction pathway in C–N coupling, in which the key coupling N- and C-intermediates are *NH₂ and *CO, respectively. 3. M/FeNi(OH)₂ interface plays an important role in facilitating H-OH dissociation and the subsequent deoxyreduction process, which improves the formation kinetics of *CO and *NH₂. As such, urea electrosynthesis is boosted. 4. An greatly enhanced urea yield rate (436.9 mmol g_{cat.}⁻¹ h⁻¹) and urea FE (66.4%), as well as ultra-long cycling stability of 1000 h are achieved, which is far exceeding than that in the mentioned paper. In addition, the

provided literature has been added in Ref. 47 in the revised manuscript.

21. In the literature review table, please include the urea current densities as well.

Reply: We thank the referee for his/her helpful suggestion. According to the suggestion, the urea partial current density (j_{urea}) has been added in Supplementary Table 3. in the revised supporting information.

Minor Comments

1. Line 10 has some errors which makes it hard to understand.

Reply: We thank the referee for his/her careful suggestion. According to the suggestion, “the unmatched of kinetics in CO₂ and NO₃⁻ reduction...” has been corrected to “the unmatched kinetics in CO₂ and NO₃⁻ reduction...” in the revised manuscript.

2. The spelling of conclusion is wrong in line 412.

Reply: We thank the referee for his/her careful suggestion. According to the suggestion, we have corrected the mistake in the revised manuscript.

3. The figure quality needs improvement.

Reply: We thank the referee for his/her helpful suggestion. According to the suggestion, Fig. 1a-1c, 1h has been replaced with high quality figures in the revised manuscript.

We thank again the referee for his/her valuable suggestions to improve the quality of our work. We hope our revision have solved the concerns of the referee and reached the quality for publication on *Nature Communications*.

Reviewer #3 (Remarks to the Author):

*In this manuscript, Xu et al. studied the urea electrosynthesis from CO₂ and NO₃⁻ by using Pd₄Cu₁-Ni(OH)₂ or Pd₄Cu₁-FeNi(OH)₂ as the catalysts. A high urea yield rate of 436.9 mmol/g and FE of 66.4%, as well as ultralong cycling stability of 1000h, are achieved for the Pd₄Cu₁-FeNi(OH)₂ catalysts. They report that Cu is atomically dispersed in Pd and the Cu single-atom alloy promotes the pivotal C-N coupling between *NH₂ and *CO intermediates. The catalytic performance in this work is impressive, while the high performance of Pd₄Cu₁-FeNi(OH)₂ is not well understood. Before further consideration of the publication of this manuscript. Several comments are listed below for the authors' reference.*

Reply: We are grateful to the referee for his/her pertinent evaluation of our work, and appreciate his/her suggestions to help us further improve the quality of our manuscript. The mechanism of the greatly enhanced urea electrosynthesis performance for Pd₄Cu₁-FeNi(OH)₂ has been revealed, and the details can be seen in the response to question 1 below.

1. The urea yield rate is significantly improved from 60.4 mmol/g to 436.9 mmol/g by Fe-doping in Ni(OH)₂, which supports the Pd₄Cu₁ nanoparticles for catalyzing C-N coupling. However, the reason for this improvement remains unclear and has not been well studied in this manuscript. Most of the manuscript focuses on the catalyst without doping, which has relatively inferior performance.

Reply: We thank the referee for bringing this to our consideration. To further uncover the mechanism of Fe doping in Ni(OH)₂ nanosheets on the greatly enhanced urea production, we firstly excluded the possibility of urea formation on FeNi(OH)₂ nanosheets. As shown in Fig. R21a, FeNi(OH)₂ nanosheets deliver negligible urea yield rates, suggesting that the greatly enhanced urea yield rate is not directly derived from FeNi(OH)₂ nanosheets. To further reveal whether FeNi(OH)₂ nanosheets promote the half-reactions, sole CO₂RR and NO₃RR were carried out. As shown in Fig. R21b, no reduction product, e.g., CO in CO₂RR is detected both for Ni(OH)₂ and FeNi(OH)₂ nanosheets, suggesting that Ni(OH)₂ and FeNi(OH)₂ nanosheets are inert for CO₂RR. Then, solo NO₃RR performance was evaluated, NH₃ yield rate and NH₃ FE at -0.5 V are increased from 5.6 mmol g⁻¹ h⁻¹ and 10.9% for Ni(OH)₂ nanosheets to 9.2 mmol g⁻¹ h⁻¹ and 18.0% for FeNi(OH)₂ nanosheets, indicating that Fe doping in Ni(OH)₂ nanosheets facilitate the deep reduction of NO₃⁻ to NH₃ (*Nanoscale*, **2023**, *15*, 204-214). Notably, NH₃ yield rate and NH₃ FE in solo NO₃RR are much lower than that of Pd₄Cu₁-Ni(OH)₂ (171.0 mmol g⁻¹ h⁻¹, 64.9%), As such, the produced more *NH₂ on FeNi(OH)₂ carrier in Pd₄Cu₁/FeNi(OH)₂ interface partly facilitate C-N coupling for urea formation.

The dissociation process of H-OH to produce active H atoms is extremely important in deoxyreduction processes, i.e., CO₂ + 2*H → *CO + H₂O and NO₃⁻ + 8*H →

*NH₂ + 3H₂O, which is usually overlooked. Previous reports have confirmed M/Ni(OH)₂ (M= Pt, Ru) interface could facilitate the water dissociation by forming M···H-O²⁻···Ni²⁺ interaction (*Science*, **2011**, *334*, 1256-1260, *Nat. Mater.*, **2012**, *11*, 550-557). And high-valence state Fe³⁺ doping in Ni(OH)₂ further accelerates this process by forming stronger Fe³⁺···O²⁻-H interaction (*J. Alloys Comp.*, **2020**, *823*, 153790). To support this conclusion, we also carried hydrogen evolution reaction on Pd₄Cu₁-FeNi(OH)₂ and Pd₄Cu₁-Ni(OH)₂. As shown in Fig. R22a, linear sweep voltammetry (LSV) curves indicate that the current density is greatly enhanced for Pd₄Cu₁-FeNi(OH)₂ than Pd₄Cu₁-Ni(OH)₂. The kinetics of hydrogen evolution reaction is also greatly enhanced for Fe doping in Ni(OH)₂ nanosheets (Fig. R22b). The result indicate that Fe³⁺ doping in Ni(OH)₂ carrier indeed promotes the dissociation of H-OH to produce more active H atoms for subsequent deoxyreduction process.

Fig. R21 | The comparison of FeNi(OH)₂ and Ni(OH)₂ nanosheets in (a) urea production, (b) solo NO₃RR and (c) CO₂RR. (d) Production rates of CO and NH₃ in solo CO₂RR and NO₃RR.

To further confirm that the greatly enhanced water dissociation is responsible for the greatly enhanced urea production for Pd₄Cu₁-FeNi(OH)₂, we further carried out electrochemical C–N coupling using D₂O as H-source to replace H₂O. The dissociation rate of D-OD and D transfer process are slower than that of H-OH due to isotope effect, which results in declined urea yield rate (*Angew. Chem. Int. Ed.*, **2020**, *59*, 21170-21175). As shown in Fig. R23, urea yield rate and urea FE for Pd₄Cu₁-FeNi(OH)₂ are synchronously declined (34.1 mmol g⁻¹ h⁻¹, 59.3%) to 5.7 mmol g⁻¹ h⁻¹ and 38.0% when using D₂O as H-source. Urea yield rate is about one six with D₂O as H-source than that with H₂O. Therefore, we can conclude that the

enhanced water dissociation to produce active H atoms is response for the enhanced urea yield rate for Pd₄Cu₁-FeNi(OH)₂.

Therefore, the reason for the greatly enhanced urea electrosynthesis performance after Fe³⁺ doping in Ni(OH)₂ carrier for Pd₄Cu₁-FeNi(OH)₂ can be assigned to the following: 1. Fe³⁺ doping in Ni(OH)₂ facilitates water dissociation to produce more active H atoms on Pd₄Cu₁ clusters, which enhance CO₂RR and NO₃RR (Fig. R21d). As such, more *CO and *NH₂ are formed to boost urea production. 2. More *NH₂ on FeNi(OH)₂ nanosheets itself facilitates C–N coupling in Pd₄Cu₁/FeNi(OH)₂ interface. According to the suggestion, the role of Fe-doped Ni(OH)₂ on the greatly enhanced urea formation has been highlighted in the revised manuscript and the newly obtained data has been added in Fig. 5b, 5c in the revised manuscript and supplementary Fig. 44, 45 in the revised supporting information.

Fig. R22 | (a) LSV curves and (b) the corresponding Tafel plots of Pd₄Cu₁-Ni(OH)₂ and Pd₄Cu₁-FeNi(OH)₂ composite samples.

Fig. R23 | Urea yield rate and FE for Pd₄Cu₁-FeNi(OH)₂ at -0.5 V using H₂O or D₂O as H-source.

2. The authors studied a series of Pd: Cu ratios and found that 4:1 is the best one with atomically dispersed Cu. Actually, in this case, the content of Cu is quite high (~ 20%). How to form Cu single sites at such a high-level Cu concentration. In addition, in their DFT calculations, the model is a doping Cu atom on the Pd surface, in which the Cu content is much less than the one in experiments. What's the actual structure of Pd₄Cu₁ in experiments?

Reply: We thank the referee for his/her insightful question. The maximal theoretical atomic percent of single-atom Cu in PdCu intermetallic compound is 50% (*Nat. Catal.*, **2022**, 5, 251-258). It is possible to obtain Cu single-atom alloy with Cu content reaching 20% in Pd₄Cu₁ clusters. In addition, Cu K-edge EXAFS spectrum (Fig. 1f) for Pd₄Cu₁-Ni(OH)₂ indicates the absence of Cu–Cu bond and resolves Pd–Cu bond, confirming the formation of Cu single-atom alloy structure.

In fact, there are thousands of configurations for Pd₄Cu₁ clusters. Our XAFS results can only determine single-atom Cu dispersion in Pd host, and the actual configuration is unclear. According to the suggestion, we also performed DFT calculations on a possible Pd₄Cu₁(111) surface for a comparison. As shown in Fig. R24 and Table R3, the ΔG for the key steps of *NO \rightarrow *HNO and *NH₂ \rightarrow *NH₂CO on Pd₄Cu₁(111) planes are 0.73 and 0.25 eV, comparable with that on Cu₁Pd(111) (0.74, 0.07 eV). The result indicated that doping Cu atoms in Pd lattice really facilitates the deep reduction of NO₃RR and urea formation. To simplify the investigation, Cu single-atom in Pd host was employed as a model in the manuscript.

Fig. R24 | Energy profiles of each elementary step in NO₃RR with C–N coupling toward urea synthesis catalyzed by Cu₁Pd(111) and Pd₄Cu₁(111) planes.

Table R3. The Gibbs free energy change (ΔG , eV) of reaction for urea formation on Cu₁Pd(111) and Pd₄Cu₁(111) at 0 V (vs. RHE).

Intermediates	Cu ₁ Pd(111)	Pd ₄ Cu ₁ (111)
*NO ₂	0	0
*NO ₂ H	0.39	0.45
*NO	-1.87	-1.89
*HNO	-1.13	-1.16
*H ₂ NO	-1.37	-1.66
*H ₂ NOH	-1.56	-1.92
*NH ₂	-3.44	-3.42
*NH ₂ CO	-3.37	-3.17
*NH ₂ CONH ₂	-9.81	-9.69
*+NH ₂ CONH ₂	-10.36	-10.34

3. In Table 1, Urea can be produced when using HCOOH as the C-source. What is the reaction pathway? How to form *CO for C-N coupling when using HCOOH as the C-source?

Reply: We thank the referee for his/her insightful question. Urea yield rate was 1.8 mmol g⁻¹ h⁻¹ when HCOOH was used as C-source at -0.5 V, much lower than that with CO₂. Previous report (*J. Photochem. Photobiol. B Biol.*, **2014**, 152, 43-46) has confirmed that HCOOH can be thermally decomposed to CO (HCOOH → CO + H₂O) through a local heating in electroreduction process (e.g., CO₂RR). To confirm the possibility, electrolysis was performed at -0.5 V in the presence of 0.1 M HCOOH + 0.1 KOH. As shown in Fig. R25, bits of CO (at 1.4 min) is detected in the electrolytic tank, confirming the decomposition of HCOOH to CO under the reduction potential. Therefore, the possible reaction pathway is listed as follow:

Fig. R25 | GC spectrum of the gas product in the electrolysis of HCOOH at -0.5 V using Pd₄Cu₁-Ni(OH)₂ as a catalyst.

4. Lines 349-351, in the DFT calculations, the differential charge density is used to confirm that "*NH₂ prefers to adsorb Cu sites which *CO on Pd sites". How to quantitatively estimate the adsorb strength from differential charge density? Why not use energy to confirm the adsorb site of *CO and *NH₂?

Reply: We thank the referee for his/her meaningful suggestion. The differential charge density in Fig. 4a and 4b can more intuitively show the tendency of electron transfer between the intermediates of *CO and *NH₂ and the active sites, which determine the strength of interaction. According to the suggestion, we also calculated the adsorption energy, the results indicate that *NH₂ possesses a larger adsorption

energy (-2.59 eV) on Cu sites than that of *CO (-2.16 eV). The results indicate that *NH_2 prefers to adsorb on Cu sites while *CO on Pd sites. According to the suggestion, the newly added result has been added in the revised manuscript.

5. In Figure 4d, the *NH_3 formation is energetically more favorable than *NH_2CO (C-N coupling). It seems to contradict the experimental results.

Reply: We thank the referee for bringing this to our consideration. In Fig. 4d, the ΔG of $^*NH_2 \rightarrow ^*NH_3$ in NO_3RR is -0.91 eV, while $^*NH_2 + ^*CO \rightarrow ^*CONH_2$ process in urea formation is endothermic reaction with energy barrier of 0.08 eV. We totally agree that it seems that *NH_3 formation is energetically more favorable than *NH_2CO . In fact, the second C–N coupling in urea synthesis releases energy of 6.45 eV, which can compensate the first C–N coupling process. As shown in Fig. R26a, the LSV curves indicate that the current density is greatly declined in the coelectrolysis of $NO_3^- + CO_2$ than that of NO_3RR . The result indicates that NO_3RR is greatly affected by CO_2RR . A possible speculation is that CO_2 is adsorbed on the catalyst surface and then converts to *COOH , which makes the catalyst surface acidic. As is well-known, NO_3RR activity is pH-dependent and the activity is greatly declined in acidic microenvironment (*J. Am. Chem. Soc.*, **2020**, *142*, 7036-7046). To confirm this speculation, we further tuned the pH of the electrolyte from 8.2 to 10 to assess C–N coupling. As shown in Fig. R26b, urea yield rate is slightly increased to 20.5 $mmol\ g^{-1}\ h^{-1}$ at -0.5 V, suggesting that the occurrence of CO_2RR on the catalyst surface really inhibits NO_3RR , which well explains high urea yield rate and urea FE for $Pd_4Cu_1-Ni(OH)_2$ in coelectrolysis of CO_2 and NO_3^- .

Fig. R26 | (a) LSV curves of $Pd_4Cu_1-Ni(OH)_2$ recorded in the mixture of $KHCO_3 + KNO_3$ under CO_2 flow in reference with that in KNO_3 , $KHCO_3 + CO_2$, $KNO_3 + KHCO_3$. (b) Urea yield rates and urea FEs assessed in the pH of 8.2 or 10 for $Pd_4Cu_1-Ni(OH)_2$ at -0.5 V.

We thank again the referee for his/her valuable suggestions to improve the quality of our work. We hope our revision have solved the concerns of the referee and reached the quality for publication on *Nature Communications*.

REVIEWER COMMENTS

Reviewer #1 (Remarks to the Author):

The authors documented that the Cu²⁺ underpotential deposition is the main reason for the synthesis of CuPd single atom alloy. The cited references (Chem. Rev., 2016, 116, 10414-10472, Electrochimica Acta, 2017, 229, 415-421) do not support the assumption. It still remains unclear of the formation of the SAA. Besides, the authors should check whether the dominated WT signal is exclusively ascribed to Pt-Cu scattering in Fig. R2d. With respect to the catalytic mechanism, the effect of the support, specifically Ni(OH)₂ and FeNi(OH)₂, is still muddled. The authors conducted the urea synthesis over the bare support. However, the authors ignore the interaction between Pd₄Cu and the support. The authors declare that Pd₄Cu₁-FeNi(OH)₂ facilitate hydrogen evolution for urea formation, why not the competitive HER? The function of the synergy or the interfacial interaction within the catalyst (Pd₄Cu and support) on urea formation are not discussed in both experimental and DFT results. The revised manuscript present still cannot address the issues. I cannot recommend the publication in Nature Communication.

Reviewer #2 (Remarks to the Author):

Although the authors have addressed the raised concerns in some capacity, I still have reservations regarding the substantial novelty of this work since the PdCu system has already been reported to yield urea. Also, the urea current density and FE are significantly lower than what has been shown for other simpler catalysts. For example, Cu (<https://pubs.rsc.org/en/content/articlelanding/2023/ee/d3ee00008g>) and Ag (<https://chemrxiv.org/engage/api-gateway/chemrxiv/assets/orp/resource/item/641b5c08aad2a62ca12f00d2/original/discovery-of-ag-as-an-active-and-selective-catalyst-for-the-electrochemical-synthesis-of-urea-from-no3-and-co2-with-100-selectivity-at-100-m-a-cm2-urea-current-density.pdf>)

Reviewer #3 (Remarks to the Author):

The authors addressed my concerns, and I have no more comments.

Response to Comments

Reviewer #1 (Remarks to the Author):

1. *The authors documented that the Cu²⁺ underpotential deposition is the main reason for the synthesis of CuPd single atom alloy. The cited references (Chem. Rev., 2016, 116, 10414-10472, Electrochimica Acta, 2017, 229, 415-421) do not support the assumption. It still remains unclear of the formation of the SAA.*

Reply: We thank the referee for his/her meaningful question. Our control experiments (see previous response to question 1 of reviewer #1) have confirmed that isolated Cu nanoparticles could not be formed due to lower reduction potential of Cu²⁺/Cu⁰ (0.340 V) than that of Pd²⁺/Pd⁰ (0.951 V). Therefore, Cu²⁺ underpotential deposition on Pd ensures the formation of PdCu alloy, instead of isolated Cu nanoparticles. It should be emphasized that Cu²⁺ underpotential deposition is not the reason for the formation of Cu single-atom alloy. The possible reason for the formation of Cu single-atom alloy may be attributed to the fast reduction kinetics of PdCl₄²⁻ ions. In our synthesis, NaBH₄ (reducing agent) has strong reduction ability which can drive fast reduction of PdCl₄²⁻ to Pd⁰. However, Cu²⁺ underpotential deposition on Pd surface is driven by thermodynamics, which typically delivers slow kinetics. That is to say, when a Cu atom bonds to Pd surface, more Pd atoms are quickly reduced and wrap around Cu atom, which inhibits the formation of Cu–Cu bond. As such, Cu single-atom was formed. More importantly, XAFS results (Fig. 1f, Supplementary Table 2) reveal the absence of Cu–Cu bond, and Pd–Cu bond is resolved in Pd₄Cu₁-Ni(OH)₂ composite sample, confirming Cu single-atom alloy feature. Of course, the detailed investigation of the real mechanism for the formation of Cu single-atom alloy is on the way.

2. *Besides, the authors should check whether the dominated WT signal is exclusively ascribed to Pt-Cu scattering in Fig. R2d.*

Reply: We thank the referee for bringing this to our attention. The wave vector numbers (abscissa value) for Pd₄Cu₁ and Pd foil are too close each other, and it is strictly impossible to completely distinguish between Pd–Cu and Pd–Pd scattering in Supplementary Fig. 9d. According to the suggestion, we have corrected “Pd–Cu scattering” to “Pd–Cu or Pd–Pd scattering” in Supplementary Fig. 9d in the revised supporting information. It should be emphasized that the revision does not affect the judgment of Cu single-atom alloy structure.

3. *With respect to the catalytic mechanism, the effect of the support, specifically Ni(OH)₂ and FeNi(OH)₂, is still muddled. The authors conducted the urea synthesis over the bare support. However, the authors ignore the interaction between Pd₄Cu and the support. The authors declare that Pd₄Cu₁-FeNi(OH)₂ facilitate hydrogen evolution for urea formation, why not the competitive HER? The function of the*

synergy or the interfacial interaction within the catalyst (Pd₄Cu and support) on urea formation are not discussed in both experimental and DFT results.

Reply: We thank the referee for his/her insightful question. The synergistic effect of Pd₄Cu₁ clusters and Ni(OH)₂ or FeNi(OH)₂ carriers can be ascribed to the following three possible factors: 1. Charge transfer between Pd₄Cu₁ clusters and Ni(OH)₂ or FeNi(OH)₂ carriers. 2. Dual active sites for C–N coupling on Pd₄Cu₁/Ni(OH)₂ interface. 3. The enhanced H–OH dissociation kinetics to boost the formation of active *H atoms. As shown in Fig. 1d, the binding energy of Cu 2p_{3/2} is slightly shifted from 932.3 eV for Cu-Ni(OH)₂ sample to higher value of 932.6 eV for Pd₄Cu₁-Ni(OH)₂. The result indicates the charge transfer is occurred between Cu and Pd atoms. Pd 3d spectrum in Supplementary Fig. 7c and Pd K-edge XANES spectra of Pd₄Cu₁ in Supplementary Fig. 9a all confirm the metallic feature of Pd without denoting or accumulating electrons on Pd. The results indicate that the charge transfer between Pd₄Cu₁ clusters and Ni(OH)₂ carriers can be ignored for Pd₄Cu₁-Ni(OH)₂. This conclusion can be further confirmed by the Pd₄Cu₁-XC-72 and Pd₄Cu₁-rGO cases in Supplementary Fig. 41. Similar with Ni(OH)₂ carrier, XC-72 and rGO carriers all possess smaller work function than that of Pd and Cu (4.65 eV for Cu, 5.14 eV for Pd{100}, 3.70 eV for Ni(OH)₂, *J. Mater. Sci. Tech.*, **2020**, 58, 73–79, *Angew. Chem. Int. Ed.*, **2014**, 53, 12120–12124). However, Pd₄Cu₁-XC-72 (1.4 mmol g_{cat.}⁻¹ h⁻¹, 6.1%) and Pd₄Cu₁-rGO (3.0 mmol g_{cat.}⁻¹ h⁻¹, 1.0%) composite samples all deliver much lower urea yield rates and urea FEs than that of Pd₄Cu₁-Ni(OH)₂ sample (18.8 mmol g_{cat.}⁻¹ h⁻¹, 76.2%). Therefore, the possible charge transfer between Pd₄Cu₁ clusters and Ni(OH)₂ or FeNi(OH)₂ carriers on the interface is not responsible for the enhanced urea formation.

As Ni(OH)₂ carriers are inert for CO₂RR and have certain capacity to catalyze NO₃RR (Supplementary Fig. S48), we infer whether there is a possibility that the produced *NH₂ adsorbed on Ni(OH)₂ or FeNi(OH)₂ can be coupled to adjacent *CO adsorbed on Pd₄Cu₁ surface on the Pd₄Cu₁/Ni(OH)₂ interface. Therefore, theoretical calculation was carried out to investigate the energy barrier of the first C–N coupling in urea formation. As shown in Fig. R1, R2, the energy barriers are 0.50 and 0.27 eV for Cu₁Pd/Ni(OH)₂ and Cu₁Pd/FeNi(OH)₂ interface, respectively, higher than that on Cu₁Pd surface (0.07 eV). The result indicates that C–N coupling tends to occur on Pd₄Cu₁ surface, instead of Pd₄Cu₁/Ni(OH)₂ or Pd₄Cu₁/FeNi(OH)₂ interface. In this case, the enhanced *NH₂ formation kinetics on Fe-doped Ni(OH)₂ carrier will not affect urea yield rate (Fig. 5g, Supplementary Fig. 48). But it is obviously not consistent with the fact of the greatly enhanced urea yield rate for Pd₄Cu₁-FeNi(OH)₂ (63.5 mmol g_{cat.}⁻¹ h⁻¹, 59.7%) than Pd₄Cu₁-Ni(OH)₂ (18.8 mmol g_{cat.}⁻¹ h⁻¹, 76.2%). In other words, dual active sites for C–N coupling on Pd₄Cu₁/Ni(OH)₂ interface is insignificant.

Fig. R1. | Adsorption configurations of the first C–N coupling of *NH₂ and *CO to form *CONH₂ on (a,b) Cu₁Pd surface, (c,d) Cu₁Pd/Ni(OH)₂ and (e,f) Cu₁Pd/FeNi(OH)₂ interface.

Fig. R2. | The energy barriers of the first C–N coupling of *NH₂ and *CO to form *CONH₂ on Cu₁Pd surface, Cu₁Pd/Ni(OH)₂ and Cu₁Pd/FeNi(OH)₂ interface.

To further confirm the enhanced dissociation of H–OH bond assisted by Pd₄Cu₁/Ni(OH)₂ interface to produce active H atoms on Pd₄Cu₁ surface is response for the greatly enhanced urea yield rate, theoretical calculations were carried out. As shown in Fig. R3a, R3b, water molecule tends to bond on Ni(OH)₂ and FeNi(OH)₂ surface on Pd₄Cu₁/Ni(OH)₂ interface by forming Ni^{δ+}⋯O²⁻–H interaction (*Science*, **2011**, 334, 1256–1260, *Nat. Mater.*, **2012**, 11, 550). As such, H–OH bond is stretched, which facilitates the dissociation of H–OH bond to form more active H atoms on Pd₄Cu₁ surface. This conclusion is confirmed by the greatly declined dissociation energy of H–OH bond, in which the energy barriers are –0.25 and –0.27 eV on Cu₁Pd/Ni(OH)₂ Cu₁Pd/FeNi(OH)₂ interface, respectively, much lower than that on Cu₁Pd surface (0.27 eV). This result also confirms that Fe³⁺ doping in Ni(OH)₂ carrier further declines the dissociation energy of H–OH to boost the formation of active H atoms. The results are consistent with enhanced hydrogen evolution activity for Pd₄Cu₁-FeNi(OH)₂ (Supplementary Fig. 47) and isotope experiment (Fig. 5e). Then, infrared spectroscopy characterization was carried out to confirm the unique adsorption configuration of H₂O molecule on Pd₄Cu₁/Ni(OH)₂ interface. Fig. R4 shows the infrared spectra of absorbed trace water on KBr, Pd₄Cu₁-Ni(OH)₂, Pd₄Cu₁-FeNi(OH)₂ composite samples. The vibration signal located at around 1640 cm⁻¹ is ascribed to bending mode of H₂O (*iScience*, **2022**, 25, 104835). The peak is slightly shifted from 1636.9 cm⁻¹ for individual H₂O to 1647.2 cm⁻¹ for Pd₄Cu₁-FeNi(OH)₂ composite sample. That is because Ni^{δ+}⋯O²⁻–H⋯Pd₄Cu₁ interaction for adsorbed H₂O molecule on the interface hinders the bending of H₂O, confirming the unique adsorption of H₂O.

Fig. R3. | Adsorption configurations of H₂O on (a) Cu₁Pd/Ni(OH)₂ and (b) Cu₁Pd/FeNi(OH)₂ interface. (c) The energy barriers of the dissociation of H–OH bond on Cu₁Pd surface, Cu₁Pd/Ni(OH)₂ and Cu₁Pd/FeNi(OH)₂ interface.

Fig. R4 | Fourier transform infrared spectra of Pd₄Cu₁-Ni(OH)₂, Pd₄Cu₁-FeNi(OH)₂ composite sample and KBr after trace water adsorption.

To further reveal the boosted *H atoms on Pd₄Cu₁ surface facilitate NO₃RR and CO₂RR, instead of H–H coupling to release H₂, we also calculated energy barriers of H–H coupling and hydrogenation of *NO₃ and *CO₂. As shown in Fig. R5, R6, the energy barrier is 0.02 eV for *NO₃ + *H → *HNO₃ process on Cu₁Pd surface in the co-existence of *H and *NO₃, much lower than that of *H → 1/2H₂ process (0.23 eV). The result indicates that the produced active H atoms on Pd₄Cu₁ surface tend to add to adjacent *NO₃, instead of H–H coupling to release H₂. Similarly, active *H atoms also tend to add to adjacent *CO₂ to boost CO₂RR in the co-existence of *H and *CO₂. Especially for Pd₄Cu₁-FeNi(OH)₂, Fe³⁺ doping in Ni(OH)₂ carrier further enhances water splitting to boost *H formation kinetics. As such, the formation kinetics of *NH₂ and *CO are boosted, and then the following C–N coupling toward urea formation is also boosted. This conclusion is verified by the greatly enhanced NH₃ and CO yield rates in solo NO₃RR and CO₂RR for Pd₄Cu₁-FeNi(OH)₂ (Fig. 5f). It should be noted that the produced *H atoms on Pd₄Cu₁ surface can promote all the deoxyreduction processes (CO₂ → *CO, NO₃⁻ → *NH₂) in CO₂RR and NO₃RR. This result also explains the comparable urea FE for Pd₄Cu₁-Ni(OH)₂ and Pd₄Cu₁-FeNi(OH)₂ composite samples. According to the suggestion, the newly obtain data has been added in Fig. 5a-5c in the revised manuscript, Supplementary Fig. 42, 43, 46, 49 and 50 in the revised supporting information.

Fig. R5. | Adsorption configurations of (a) $*NO_3 + *H$, (b) $*HNO_3$, (c) $*CO_2 + *H$ and (d) $*COOH$ on Cu₁Pd-FeNi(OH)₂.

Fig. R6. | The energy barriers of H-H coupling to form H₂ or the hydrogenations of $*NO_3$ and $*CO_2$ to form $*HNO_3$ and $*COOH$.

4. *The revised manuscript present still cannot address the issues. I cannot recommend the publication in Nature Communication.*

Reply: We thank again the referee for his/her valuable suggestions to improve the quality of our work. We hope our revision have solved the concerns of the referee and reached the quality for publication on *Nature Communications*.

Reviewer #2 (Remarks to the Author):

1. Although the authors have addressed the raised concerns in some capacity, I still have reservations regarding the substantial novelty of this work since the PdCu system has already been reported to yield urea.

Reply: We thank to the referee for his/her question. Although the PdCu system has been reported in the coupling of CO₂ and NO₃⁻ toward urea formation (*EES Catal.*, **2023**, *1*, 45–53), our work is totally different from the reported work. The differences between two works are listed as follow: 1. The totally different catalyst system: our work reported a Cu single-atom alloy clusters anchored on FeNi(OH)₂ surface, while the mentioned work reported PdCu intermetallic compound on carbonized bacterial cellulose. 2. The totally different reaction pathways for C–N coupling: urea formation in our work undergone the C–N coupling of *NH₂ and *CO (*NH₂ + *CO → *CONH₂, *CONH₂ + *NH₂ → *CO(NH₂)₂), which was confirmed by *in situ* Raman spectroscopic characterizations. The mentioned work reported a totally different C–N coupling pathway of *NO₂ + *CO₂ → *CO₂NO₂, *CONH₂ + *NO₂ → *CONO₂NH₂. 3. The greatly enhanced urea electrosynthesis performance: our work achieves a recorded urea yield rate of 436.9 mmol g_{cat.}⁻¹ h⁻¹ and urea FE of 66.4% in GDE at –0.6 V *versus* RHE, much higher than that of the mentioned work (12.7 mmol g_{cat.}⁻¹ h⁻¹ at –0.5 V, 69.1 ± 3.8% at –0.5 V). More importantly, our Pd₄Cu₁-FeNi(OH)₂ sample delivers outstanding cycling stability ability of 1000 h, far exceeding than that of PdCu intermetallic compound (10 h). And finally 1.05 g urea was obtained, demonstrating excellent potential practical value.

Beyond that, the new findings of our work are emphasized as follow: 1. we put forward kinetics matching of C- and N-intermediates is the determining factor to achieve high urea yield rate and urea FE in electrochemical coupling of CO₂ and NO₃⁻. 2. The formation kinetics of C- and N-intermediates can be tuned by regulating Cu doping level in Cu single-atom alloy. 3. The formation kinetics of active H atoms by water splitting plays an important role in deoxyreduction process, and this process can be tuned by constructing metal/FeNi(OH)₂ interface. The finding may help the researchers to deep understanding C–N coupling process and catalyst design toward high C–N coupling performance. Moreover, the finding may be extended to other coupling processes, such as C–C coupling to polycarbon products in CO₂RR, and oxidative coupling of methane. Besides, water splitting issue is a general problem in deoxyreduction process (e.g., CO₂RR) and usually is ignored, our work demonstrates the importance of water splitting on hydrogenation of small molecules and provides an efficient strategy to boost water splitting by constructing metal/hydroxide interface. The new viewpoint we put forward in this work makes the mechanism of urea electrosynthesis more comprehensive and profound.

2. Also, the urea current density and FE are significantly lower than what has been shown for other simpler catalysts. For example, Cu

(<https://pubs.rsc.org/en/content/articlelanding/2023/ee/d3ee00008g>) and Ag (<https://chemrxiv.org/engage/api-gateway/chemrxiv/assets/orp/resource/item/641b5c08aad2a62ca12f00d2/original/discovery-of-ag-as-an-active-and-selective-catalyst-for-the-electrochemical-synthesis-of-urea-from-no3-and-co2-with-100-selectivity-at-100-m-a-cm2-urea-current-density.pdf>).

Reply: We thank to the referee for his/her evaluation of our work. For urea partial current density, the urea current density for Cu in provided reference 1 can be deduced to be 26.87 mA cm⁻², not 115.25 mA cm⁻² (-0.41 V vs. RHE, see the calculation process below). The optimal urea current density for Pd₄Cu₁-FeNi(OH)₂ in our work is 7.05 mA cm⁻² with catalyst loading of 0.025 mg cm⁻². It is generally accepted that current density in electrolysis is determined by the catalyst loading amount. When normalized by catalyst loading amount, the urea current density for Pd₄Cu₁-FeNi(OH)₂ composite sample is 282.0 mA cm⁻² mg⁻¹ in our manuscript, much larger than that for Cu (53.74 mA cm⁻² mg⁻¹) in Ref. 1. Of course, we also attempted to increase the catalyst loading amount (0.5 mg cm⁻²) to increase urea current density. However, the catalyst was agglomerated to a dense thick member due to low density of our catalyst, which hinders the greatly enhanced urea current density.

The urea current density is claimed to reach 100 mA cm⁻² for Ag electrode in GDE in Ref. 2, however, it is missing the key parameters including catalyst loading amount, electrode area and so on. Furthermore, we notice the Ref. 2 is a preprint, not an officially published paper with rigorous peer review. Therefore, it is less rigorous to make a comparison of urea current density in our manuscript with that of Ag electrode in Ref. 2.

The specific calculation process for Cu electrocatalyst in Ref. 1 is shown as follows:

Urea yield rate: 7541.9 μg h⁻¹ mg_{cat.}⁻¹ (i.e., 125.6 mmol h⁻¹ g_{cat.}⁻¹),

Catalyst loading: 0.5 mg cm⁻²,

Electrolytic time: 1.5 h,

The area of the working electrode has not been included in the paper, therefore, the area is set to be x cm².

The produced urea amount:

$$125.6/1000 \times 1.5 \times 0.5/1000 \times x = 9.42x \times 10^{-5} \text{ mol} \quad (1)$$

According to the equation 2, to produce 1 mol urea, it is needed 16 mol electrons, the consumed electric quantity for urea:

$$9.42x \times 10^{-5} \times 16 \times 6.02 \times 10^{23} \times 1.6 \times 10^{-19} = 145.1xC \quad (3)$$

Therefore, the average urea partial current density:

$$145.1x/5400/x = 0.0268 \text{ A cm}^{-2} = 26.87 \text{ mA cm}^{-2} \quad (4)$$

We thank again the referee for his/her valuable suggestions to improve the quality of our work. We hope our revision have solved the concerns of the referee and reached the quality for publication on *Nature Communications*.

Reviewer #3 (Remarks to the Author):

The authors addressed my concerns, and I have no more comments.

Reply: We thank again the referee for his/her valuable suggestions to improve the quality of our work.

REVIEWERS' COMMENTS

Reviewer #1 (Remarks to the Author):

The authors have addressed the issues. Now, this revised manuscript can be accepted.